# Mining mutation contexts across the cancer genome to map tumor site of origin

Saptarshi Chakraborty[1], Axel Martin[2], Zoe Guan [2], Colin B. Begg [2✉] & Ronglai Shen [2✉]

The vast preponderance of somatic mutations in a typical cancer are either extremely rare or have never been previously recorded in available databases that track somatic mutations. These constitute a hidden genome that contrasts the relatively small number of mutations that occur frequently, the properties of which have been studied in depth. Here we demonstrate that this hidden genome contains much more accurate information than common mutations for the purpose of identifying the site of origin of primary cancers in settings where this is unknown. We accomplish this using a projection-based statistical method that achieves a highly effective signal condensation, by leveraging DNA sequence and epigenetic contexts using a set of meta-features that embody the mutation contexts of rare variants throughout the genome.

---

[1] Department of Biostatistics, State University of New York at Buffalo, Buffalo, NY, USA. [2] Department of Epidemiology and Biostatistics, Memorial Sloan-Kettering Cancer Center, New York, NY, USA. ✉email: beggc@mskcc.org; shenr@mskcc.org

The last decade has witnessed the explosion of information available from large-scale tumor sequencing projects, including the Cancer Genome Atlas (TCGA) project that cataloged somatic variants in over 10,000 tumor samples using whole exome sequencing[1], the more recent release of the International Cancer Genome Consortium Pan-Cancer Analysis of Whole Genomes (ICGC-PCAWG) dataset of whole genome sequencing of over 2000 tumors[2], and various targeted cancer gene sequencing panels in clinical applications such as the MSK-IMPACT effort at our own institution[3]. These large information resources offer the potential to develop statistical tools for predicting the primary site of origin of cancers using mutational profiles detected from tumor DNA, an emerging field with great practical utility for clinical diagnostics and early cancer detection[4–8]. For example, cancers of unknown primary typically occur at metastatic stages where the tumors have already metastasized and spread to other organ sites: e.g., in the diagnosis of carcinoma of unknown primary there are lesions present at multiple organ sites and the primary site is unknown. In circulating tumor DNA, fragments of tumor DNA are identified in blood and can potentially facilitate early detection of cancer.

It has been observed historically that somatic alterations in major oncogenes and tumor suppressor genes occur in a highly lineage-dependent manner (e.g., *KRAS* mutations in cancers of the pancreas, colon, and lung)[9]. More recently, Haigis et al.[10] have argued that such tissue-specificity should be in fact the rule, not the exception, because somatic alterations in a tumor must function within the framework of the transcriptional network and epigenetic states established by its developmental lineage in the corresponding tissue of origin. To that end, epigenomic features such as chromatin accessibility and histone modification patterns have been found as major determinants of the cancer mutational landscape that is specific to different cells of origin[11]. Nevertheless, existing data on epigenome organization only partially account for the mutational variation in certain cancer types. The mutational heterogeneity across the cancer genome has not been fully explored.

A major challenge in analyzing tumor mutation profiles lies in the fact that the distribution of mutation frequencies of individual variants is extremely long-tailed. That is, the vast preponderance of somatic mutations occur very rarely. This leads to very challenging problems for modeling due to the ultra-high dimension and ultra-sparse data space. Indeed Chakraborty et al.[12] reported that the preponderance of mutations in existing large-scale sequencing datasets are in fact singletons, observed only once in the cohort, posing analytical challenges similar to that encountered in estimating rare species in ecology and word frequency in computational linguistics[13,14]. One can view the current mutational landscape as an iceberg. The small proportion of the iceberg above the water surface represents the hotspots and other genetic loci where mutations have been observed to occur frequently in known cancer genes (e.g., *BRAF* V600E). This study is motivated by the belief that the vastly larger submerged portion of the iceberg, representing rare mutations and those never previously observed, contains information of substantial clinical relevance, but requires innovative statistical and computational methods to extract the signals effectively.

In this study we propose and exemplify a rigorous statistical approach that permits aggregating information from this hidden genome of rare variants by leveraging their DNA sequence and epigenetic contexts. We define a set of meta-features that employ mutation contexts such as the topological position of a variant on the chromosome, the single base substitution type in the trinucleotide context, and epigenomic features such as chromatin accessibility, and use these in a method that captures the information in the hidden genome through multidimensional projections of these features allied with a hierarchical regression model.

Through iterative model fitting we formally assess the relative importance of these meta-feature groups. Comparisons with black box machine learning approaches used previously in the literature[6,7,15] demonstrate competitive predictive performance of our approach. The key advantage of our method however pertains to its interpretation and predictor attribution: through carefully constructed hierarchical layers cancer site-specific effects of individual predictors (including each hidden genome variant and each meta-feature category) are coherently quantified. These individual effects aid identification of discriminative genomic signals, and thus may potentially reveal novel biological insights. Such quantification of granular predictor effects is virtually impossible in a black box machine learning method. We apply our approach to the ICGC-PCAWG whole-genome, TCGA whole-exome, and MSK-IMPACT targeted cancer gene panel datasets. The three datasets provide a natural "reduction" experiment, and a major contribution of our study lies in the formal assessment of how the decreasing order of genome coverage affects the accuracy of prediction of different tissue sites. In particular, our results show that highly discriminative diagnostic information exists in the noncoding regions of the genome for ovarian and prostate cancer tumors, which can be fully harnessed only in a whole genome-sequencing framework. Our study focuses primarily on somatic point mutations. Accurate detection of other types of genomic alterations in a patient sample, e.g., copy number alterations (particularly, low-level chromosomal gains and losses) require much higher tumor content than somatic mutations. In scenarios where tumor content is extremely low, which is encountered typically in a fraction of tumor samples as well as in the plasma sequencing context, somatic point mutations are typically the only reliable information that is available for diagnostic purposes. We note however that the proposed methodologies can be extended in a straightforward way to incorporate information on other genomic alterations, if available.

## Results

**Method overview**. We draw upon a context-based learning approach[16] in which the role of rare and unseen variants can be "learned" through their local genome and epigenome context: a set of quantifiable knowledge units obtained from the associated DNA sequence and epigenome contexts which we refer to as meta-features. These meta-features include various topological or functional annotations of the genome and epigenome, such as single-base substitution signatures in the trinucleotide context, regional indices mapping the topological position on the chromosome, and features of the epigenome including chromatin accessibility and histone modification that we describe in more detail in the next section. We propose a projected hidden genome approach that first finds scalar projections of the meta-feature vectors along the direction of the mutation profile. This represents a signal condensation process that maps the extremely sparse variant space (dominated by singletons and rare variants) to a meta-feature space organized by genome and epigenome context. These scalar projections, induced from a hierarchical Bayesian classification model with normalized mutation profiles as predictors, constitute local mutation densities attributable to the associated meta-features, normalized by the square-root of the total mutation burden encountered in the tumor. The underlying model describes the effects of rare or previously unobserved variants through their mutation contexts as captured by these meta-features. This permits an effective condensation of information across all variants. Instead of estimating the effects of the tens of millions of rare variants separately, we now estimate them collectively through these meta-features. As a result, the ultra-high dimensional variant space is projected onto a much lower dimensional space aggregating information from variants that share similar mutational contexts

as embodied by the meta-features. These scalar projections are then used as predictors in an efficient group-penalized[17] multinomial logistic regression model for maximum marginal a posteriori estimation within a Bayesian hierarchical classification model[18] (see Methods). The model also allows inclusion of additional individual parameters that capture the "residual effects" of a handful of hotspot variants (e.g., *BRAF* V600E) that can possess lineage-dependency not captured by the meso-scale genomic and epigenomic meta-features. Use of a group-lasso penalty aids rigorous identification of consequential regression effects at individual feature/predictor levels, e.g., individual meta features and individual variants—if a specific feature is not discriminative of at least one cancer site, its regression coefficients across all cancer sites are set to zero and consequently the feature has no effect on predicting the cancer site of a new tumor.

**Description of data sources and meta-features**. We use three publicly available cancer sequencing data sets: the PCAWG whole genome[2], the TCGA whole exome[1], and the MSK-IMPACT targeted cancer gene panel[3]. We elected to focus on ten common cancer sites that were available across all three data sets, permitting a direct comparison between the different sequencing platforms: breast, colorectal, esophageal, kidney, liver, lung, skin, ovarian, pancreatic, and prostate (see Supplementary Tables 1–3 for a detailed note on the cancer histologies considered and the corresponding sample sizes for these cancer sites). In the PCAWG whole-genome dataset, a total of 36,325,180 variants (with 35,285,233 being singletons, i.e., observed only once) were detected by a consensus mutation calling approach in 1702 tumors belonging to these ten cancer sites: an average of 19,076 somatic mutations per tumor at the whole-genome level. In the TCGA dataset, a total of 1,312,572 total variants (of which 1,246,857 are singletons) were detected in 4503 tumors from one of these ten cancer sites: an average of 319 mutations per tumor at the whole-exome level. In the MSK-IMPACT dataset, a total of 25,454 variants (with 23,180 singletons) were detected using a clinical bioinformatics pipeline in 5078 tumors belonging to these ten cancer sites: an average of 6.7 mutations per tumor. The three sequencing datasets provide a natural progression in genome coverage from targeted panel to whole-exome to whole-genome. In addition, we created two "simulated" data sets that contain exactly the same cases as the PCAWG whole genome dataset with the exception that the variants included are restricted to the variants within the TCGA whole-exome and the MSK-IMPACT targeted panel coverage, respectively. This represents a simulated progression to control for differences due to factors other than genome coverage such as difference in cancer subtype composition and other clinical and demographic patient characteristics. A total of 6 and 166 tumors out of the 1702 PCAWG tumors that did not possess any variants within the TCGA whole-exome and MSK-IMPACT targeted cancer panel coverage respectively were removed from the corresponding simulated datasets for further analysis. Site-specific sample sizes in the "full" and "simulated" subsets of the PCAWG data are provided in Supplementary Table 1.

We focus on the following five meta-features: (i) cancer gene: limited to the 604 cancer genes collectively cataloged in in the OncoKB list[19] (584 genes) and MSK-IMPACT (414 genes). (ii) SBS-96 category: the single-base substitution signatures in the trinucleotide context to which the variant is associated. SBS-96 categories are distinct from derived mutation signature categories of tumors. The SBS-96 categories are characteristics of individual variants and are automatically determined given each single-nucleotide alteration. In contrast, the derived mutation signature category is a characteristic of a tumor determined by comparing

the relative mutation burdens in the tumor attributable to these 96 SBS categories, to weights precomputed in a reference dataset[20,21]. Note that so far these reference weights have been derived only for whole genome and whole exome mutation burdens and identifying derived mutation signatures using mutations only detected in a targeted gene panel is difficult. (iii) regional index—contiguous 1 Mb regions along each chromosome on which the variants are located; produces regional densities of mutation burden, the normalized total numbers of mutations detected on the underlying sequencing platform located in each region as scalar projections. (iv) chromatin accessibility: log-log DNase I hypersensitivity profile as a global measure of chromatin accessibility per 1 Mb chromosome region. (v) histone modification: log-log total reads from ChIP-seq assay for histone marks including H3K36me3 and H3K4me1 per 1 Mb chromosome region. The epigenome data were obtained from the ENCODE[22] and the Epigenome Roadmap Study[23]. Each epigenomic meta-feature listed in (iv) and (v) was precomputed from these databases for each 1 Mb chromosome region separately for the ten cancer sites considered herein. These collectively produce a vector of 30 scores (10 scores from the ten cancer sites each for chromatin accessibility, H3K36me3 and H3K4me1) through a scalar projection of the mutation profile for each tumor. [See Methods for a detailed description on these meta-features.] In addition to meta-features (i)–(v) we also include an intercept meta-feature (vector of 1's) that produces the square-root of the total mutation burden in a tumor as a scalar projection along the direction of the mutation profile vector.

**The hidden genome reveals a near-perfect separation of tumor sites**. To visualize the amount of discriminative signal present in the hidden genome, we constructed a three-dimensional approximation of the meta-feature embedded genomes using a principal component analysis followed by a t-distributed stochastic neighbor embedding (tSNE)[24] to reduce meta-feature scalar projection of mutational profile vectors into a three-dimensional subspace. The resulting embeddings are displayed as scatterplots in Fig. 1 (an interactive html version is included in the GitHub repository containing custom software developed in this study[25]) with points color-coded according to their cancer sites. These figures show that the projections in the hidden genome approach provides a near perfect separation of tissue types at the whole-genome level. Interestingly, the pancreatic cancer samples (orange dots) formed separate clusters, with neuroendocrine histology distinctly different from adenocarcinoma histology. A recent whole-genome sequencing study of pancreas neuroendocrine tumors suggested that sporadic neuroendocrine tumors contain an elevated proportion of germline mutations in the DNA repair genes *MUTYH*, *CHEK2*, and *BRCA2*[26]. In addition, a small subset of breast cancer samples (black dots), primarily of the triple negative subtype, separated out and co-clustered with a subset of ovarian tumor samples (azure blue dots). It is well known that triple negative breast cancers and high grade serous ovarian cancers are typically characterized by *BRCA1* mutations. Taken together, these observations suggest that germline genetics may further underlie the somatic mutational heterogeneity in addition to tissue type.

The degree of separation of cancer types decreases as we restrict the data to whole-exome coverage (Fig. 1b) and more so when restricted to the targeted panel coverage (Fig. 1c). The whole exome data led to a moderate separation of tissue types and the separation at the targeted panel level is relatively poor. Such information attrition suggests that the discriminative information for tissue of origin in somatic mutation data is largely embedded in the hidden genome beyond frequently mutated cancer genes.

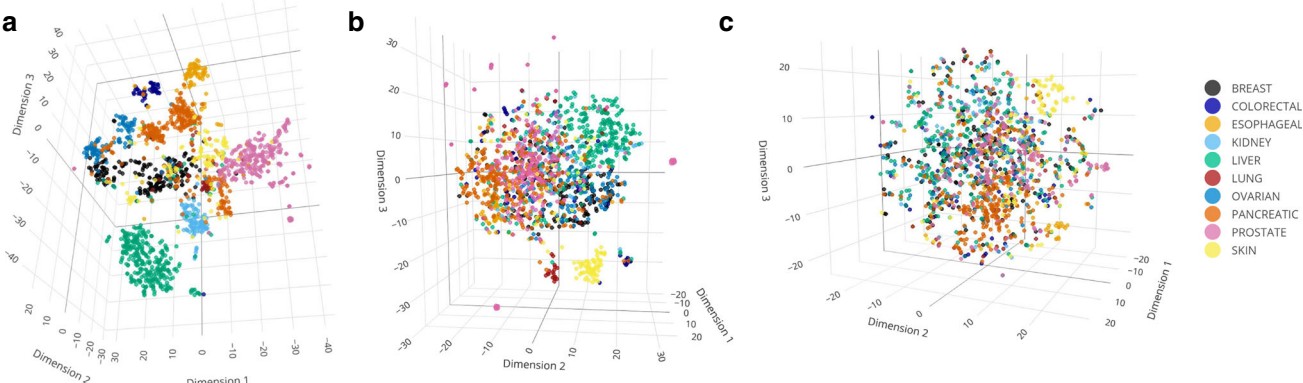

**Fig. 1 Scatter plots showing three-dimensional embeddings of meta-feature projections of ultra-high dimensional and sparse mutational profile vectors.** Each point in each scatter represents a single PCAWG tumor and is color coded according to its cancer site. For each tumor, meta-feature scalar projections along the directions of the whole genome mutational profile (**a**), simulated whole exome mutational profile (**b**), and simulated targeted sequencing mutational profile (**c**) are first obtained; 30 component principal component analysis followed by three-dimensional t-distributed stochastic neighbor embedding (t-SNE) are subsequently performed on the resulting scalar projections to obtain an approximate three-dimensional representation of each mutational profile. An interactive html version of the Figure is included in the GitHub repository containing custom software developed in this study[25].

These visualizations provide intuition for some of the more formal analytical results presented in the following sections. The impressive near disjoint clustering of these lower dimensional embeddings, particularly in the whole genome sequencing case, suggests that the proposed meta-feature based scalar-projections permit efficient and effective accumulation of discriminative signals in the ultra high-dimensional and extremely sparse mutational profiles. Consequently, a classifier utilizing these projections as predictors is expected to have a high prediction/ classification accuracy. These figures also illustrate how going from targeted cancer gene panel sequencing to whole-exome sequencing to whole-genome sequencing can substantially improve the signal, and thereby bolster the discriminative ability of the classifier. In the following sections we formally quantify these observations.

**A unified framework for integrating information across diverse sources.** To formally assess the predictive accuracy of the projected hidden genome classifier we performed ten replications of fivefold cross validation experiments separately on each of the three datasets and the two PCAWG whole-genome "simulated" datasets. The classification performance is displayed initially using precision-recall curves in Fig. 2. Here the performance of the full hidden genome model is contrasted with models that use subsets of the (meta-)features. These include a classifier based solely on individual variants (Baseline); a gene-based classifier that uses similar information to the method proposed by Soh et al.[6] (Gene); a regional mutation density classifier (RMD) and another classifier that additionally uses information on mutational signatures (RMD + SBS-96) both of which use analogous information to the methods studied by Jiao et al.;[7] and the proposed hidden genome model integrating all these features (P. Hid. Genome). Figure 2 and Supplementary Figs. 1–5 display the overall and cancer-site specific precision recall curves respectively for these classifiers across all five sequencing platforms. These figures demonstrate the ability of the projected hidden genome classifier in condensing information from diverse sources: the key advantage of the proposed approach is that it provides a unifying strategy allowing integration of all these factors through appropriate meta-feature transformations within a multinomial logistic modeling framework, which in turn permits high predictive accuracies across all DNA sequencing platforms.

**Quantifying information in the hidden genome**. The area under the precision-recall curve (AUC) provides a robust[27] quantitative summary of classification performance, with a larger area indicating better classification (an area of 1 indicates perfect accuracy). Unlike a receiver-operator characteristic (ROC) curve, a precision-recall curve adjusts for class size imbalances which regularly occur in (most) one-vs-rest comparisons obtained from a multi-class classifier. The robustness is reflected in the baseline (expected) precision-recall AUC for a null classifier which randomly assigns positive and negative class labels to sample units[27]. Figure 3 shows the macro (AUCs averaged over all sites) and the site-specific one-vs-rest precision-recall AUCs together with the corresponding null baseline AUCs (displayed as darkened areas on the bars). For each individual site-specific one-vs-rest comparison the null baseline is the relative sample size for the site[27] $\left(\text{i.e.,} \frac{\text{sample size for the site}}{\text{total sample size across all sites}}\right)$; for the "overall macro" comparison the null baseline, obtained by averaging all individual site-specific null baselines, is simply $\frac{1}{\text{number of cancer sites}}$ which is 1/10 in our study.

Our results show that meta-features introduced into the model succeed in extracting considerable amounts of new tissue specific information buried in the hidden genome. This is especially evident in the substantial increases in discriminatory accuracy obtained by moving from cancer gene panel sequencing to whole exome to whole genome data (Fig. 3a). The average one-vs-rest precision-recall AUC associated with the projected hidden genome classifier ranges from 0.59 for the cancer gene panel, to 0.75 for whole-exome and 0.90 for whole-genome. The increase is particularly prominent in ovarian and prostate cancer tumors, where the one-vs-rest AUCs range from 0.10 to 0.62 to 0.87, and 0.54 to 0.61 to 0.98, respectively. In this progression we are largely adding information from rare variants in the hidden genome, since the hotspots and known cancer genes are largely contained in the cancer gene panel sequencing data. In the whole genome dataset, the highest individual site specific AUCs are obtained in prostate and skin cancer (0.98 each), while the lowest AUC is observed in lung (0.65). The low AUC for lung in the PCAWG datasets (both real and simulated) are likely attributable to the small sample size ($n = 38$, which is further reduced to ~30 in each fivefold cross-validation training set). Figure 3b provides a comparative assessment of the classification accuracies achieved by the projected hidden genome classifier at different "simulated" sequencing coverage using the PCAWG whole-genome dataset,

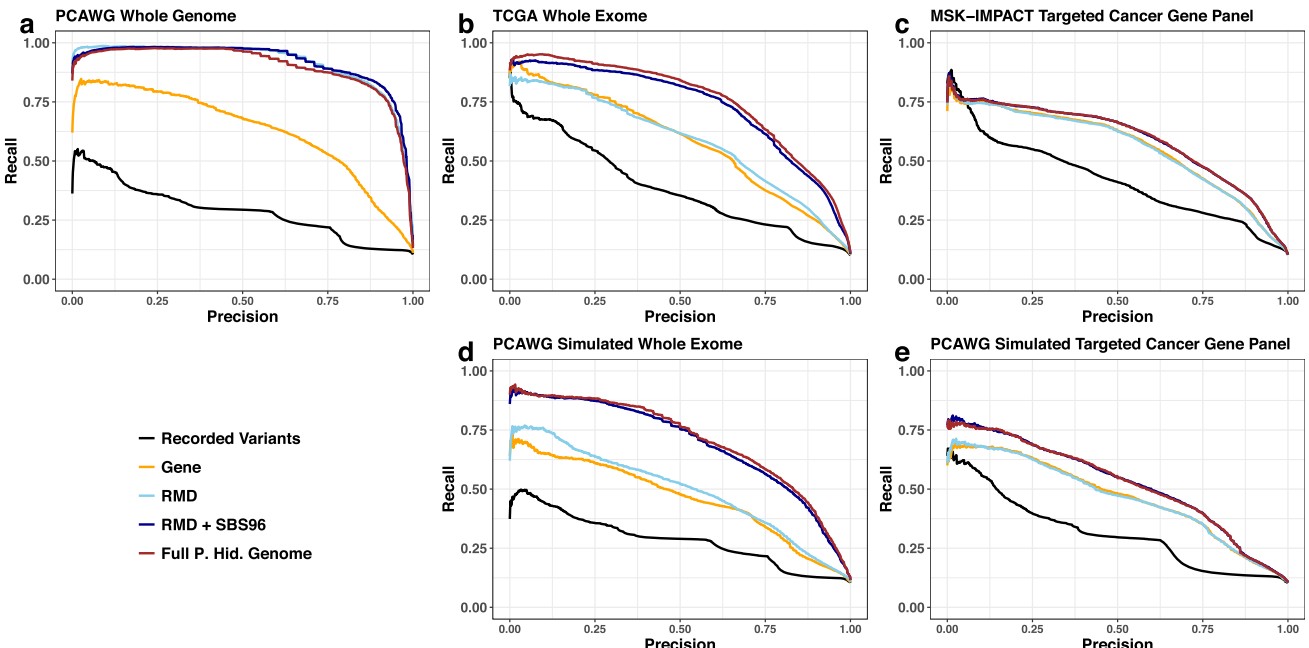

**Fig. 2 Comparing performance of multinomial logistic classifiers trained with various sets of predictors for predicting cancer site of origin.** Macro precision-recall curves comparing cross-validation predictive performances of multinomial logistic classifiers with (i) the baseline recorded variants (Baseline; black lines), (ii) cancer gene indicator (Gene; orange lines), (iii) regional mutation density (RMD; sky blue line), (iv) RMD and nucleotide change signature (RMD + SBS-96; dark blue lines), and (v) all predictors in the full projected hidden classifier (Full P. Hid. Genome; brown lines). The curves for the PCAWG whole genome, PCAWG simulated genome, and PCAWG simulated targeted panel data were displayed in Panels **a**, **b**, and **c**, respectively, while those for the TCGA whole exome data and MSK-IMPACT targeted panel data were displayed in panels **d** and **e**, respectively.

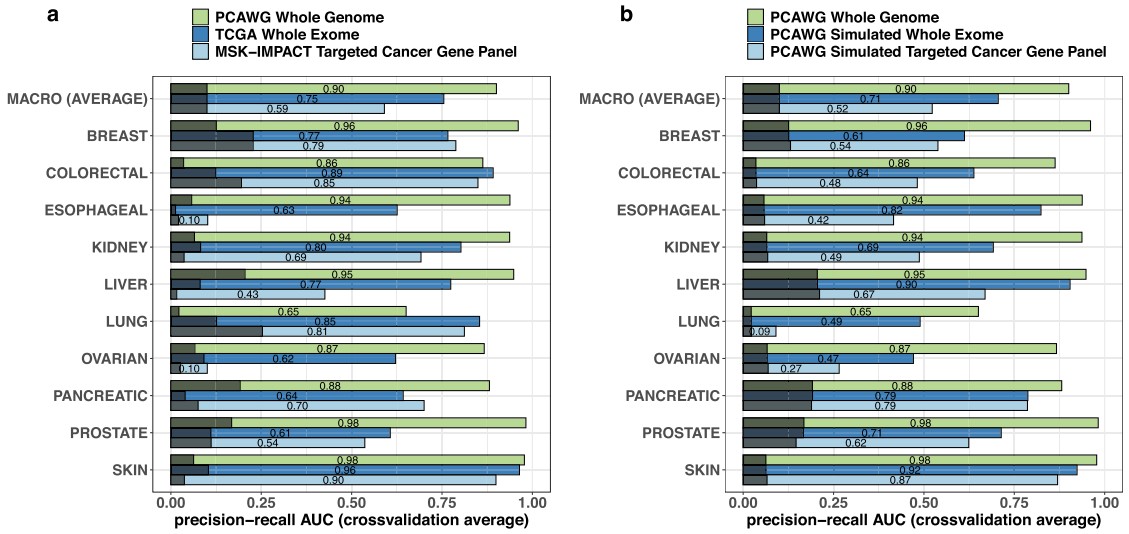

**Fig. 3 Cancer site-specific and overall predictive performances of the hidden genome classifier in different DNA sequencing datasets.** Bar charts showing classification performances of the proposed hidden genome classifier as measured by cancer site-specific and macro cross-validation precision-recall AUCs (plotted as bars), when applied to whole genome (light green), whole exome (blue), and targeted panel (light blue bars) datasets. Panel **a** shows those in the three real datasets–PCAWG genome, TCGA whole exome, and MSK-IMPACT targeted cancer gene panel, whereas Panel **b** displays the performances in the three PCAWG datasets—PCAWG genome, PCAWG simulated whole exome, and PCAWG simulated targeted panel. Overlayed on each bar, the darkened area represents the baseline null (expected) precision-recall AUC for the corresponding one-vs-rest classification associated with a classifier that randomly assigns positive and negative class labels to sample units.

showing a similar monotonic relationship between genome coverage and one-vs-rest precision recall AUC.

There are some observable differences in certain cancer type specific one-vs-rest precision recall AUCs between the simulated and the real datasets. For example, the AUCs for breast cancer are substantially higher in the TCGA exome and MSK-IMPACT targeted cancer gene panel datasets. It is also of note that the classification performance in breast cancer is marginally better in the MSK-IMPACT targeted cancer gene panel dataset than in the TCGA whole exome dataset. In contrast, AUCs for liver tumors are substantially higher in the two simulated PCAWG datasets than in the two real datasets. These differences may be attributed

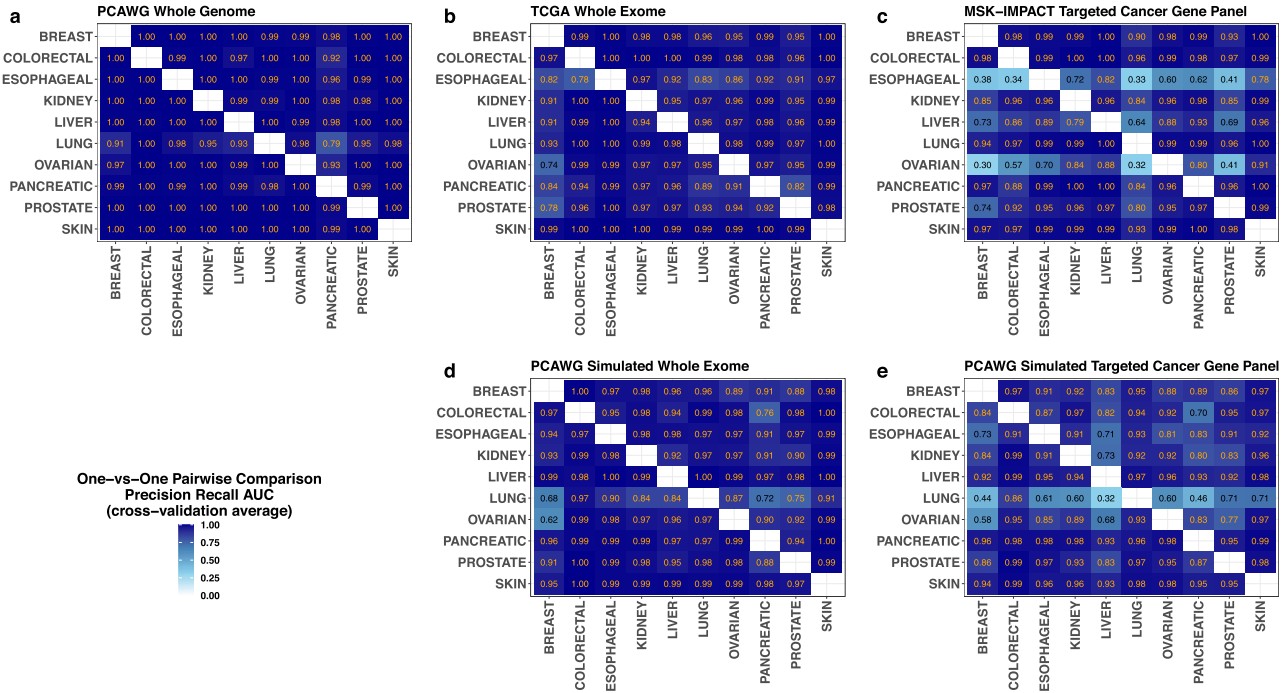

**Fig. 4 Predictive performances of the hidden genome classifier in pairwise (one-vs-one) cancer site classifications in different DNA sequencing datasets.** Heatmaps showing cross-validation precision recall AUCs for the projected hidden genome classifier in all pairwise (one-vs-one) comparisons across all real sequencing platforms, (PCAWG whole genome, TCGA whole exome, and MSK-IMPACT targeted cancer gene sequencing; panels—**a**, **b**, and **c**, respectively) and simulated sequencing platforms (PCAWG simulated whole exome and PCAWG targeted cancer gene sequencing; panels—**d** and **e**, respectively). In each heatmap, the vertical axis represents the positive class, the horizontal axis represents the negative class, and each cell displays the cross-validation precision recall AUC for the corresponding one-vs-one classification calculated from multinomial predictive probabilities provided by the projected hidden genome classifier.

to the differences in respective sample sizes which also affect the baseline null AUCs, and to the heterogeneity in patient cohorts and tumor subtype compositions in the datasets.

We also evaluated predictive performances of the classifier in pairwise (one-vs-one) cancer site classifications. Pairwise classification probabilities were obtained by conditioning the ten-site/class multinomial predicted probabilities to every class pair (see the Methods for more details). Precision recall AUCs of the corresponding paired classifications were subsequently evaluated separately for each sequencing platform. The AUCs for these ordered paired comparisons across all platforms are displayed as heatmaps in Fig. 4, with darker shadings reflecting higher diagnostic accuracies. As depicted in the figure, almost all pairwise comparisons are observed to have very high AUCs in the whole genome and the whole exome datasets (Fig. 4a, b). Even in targeted panel datasets, pairwise comparisons show similarly high discriminatory accuracies for most tumor types with exceptions in esophageal, liver, and ovarian cancers (Fig. 4c). The simulated whole-exome and targeted panel dataset show similar trends (Fig. 4d, e).

**Quantifying the marginal and cumulative predictive value of meta-feature groups.** Our model facilitates a comparative assessment of the marginal and cumulative effects of the various meta-feature groups used in the projected hidden genome classifier. Figure 5a–e display the marginal and cumulative macro one-vs-rest precision recall AUCs of the various meta-feature groups in different platforms. The bars display the marginal effects of each meta-feature group, and the solid points on the curves show the cumulative effects of each meta-feature group when the group is added to the cumulative group of meta-features to its immediate

left. Residual effects of individual common variants and an intercept are included in all models. As depicted in Fig. 5a, the most informative meta-feature group in terms of marginal effects in the whole genome dataset corresponds to the 1 Mb regional indices (macro AUC = 0.92), followed by the SBS-96 categories (macro AUC = 0.86). The cumulative effects steadily increase when genes, SBS-96 and regional indices are sequentially added to the baseline classifier and stabilize thereafter. In contrast, at the whole exome level, both simulated whole-exome (Fig. 5d) and TCGA whole-exome (Fig. 5b) SBS-96 categories appear to be the most informative meta-feature, followed by genes and regional indices which have near identical marginal effects. The cumulative effects increase when genes and SBS-96 categories are added successively to the model and stabilize thereafter. Finally, in the targeted sequencing datasets genes (Fig. 5c, e), SBS-96 categories and regional indices all appear to have similar small marginal effects (with SBS-96 further smaller in MSK-IMPACT data) as these global meta-features require higher genome coverage to be used effectively. The overall effect first increases moderately when genes and SBS-96 are successively added to the classifier and thereafter stabilizes. The epigenetic features display moderate marginal effects and do not show discernable cumulative effects when used in conjunction with the other meta-features in any of the five datasets.

**Identifying the most discriminatory individual features.** Our model permits rigorous quantification of the effects of individual variants and individual meta-features in different cancer sites. To visualize these effects we constructed odds-ratios for being classified, relative to not being classified (one-vs-rest odds ratios, see Methods) to the corresponding tissue site for one standard

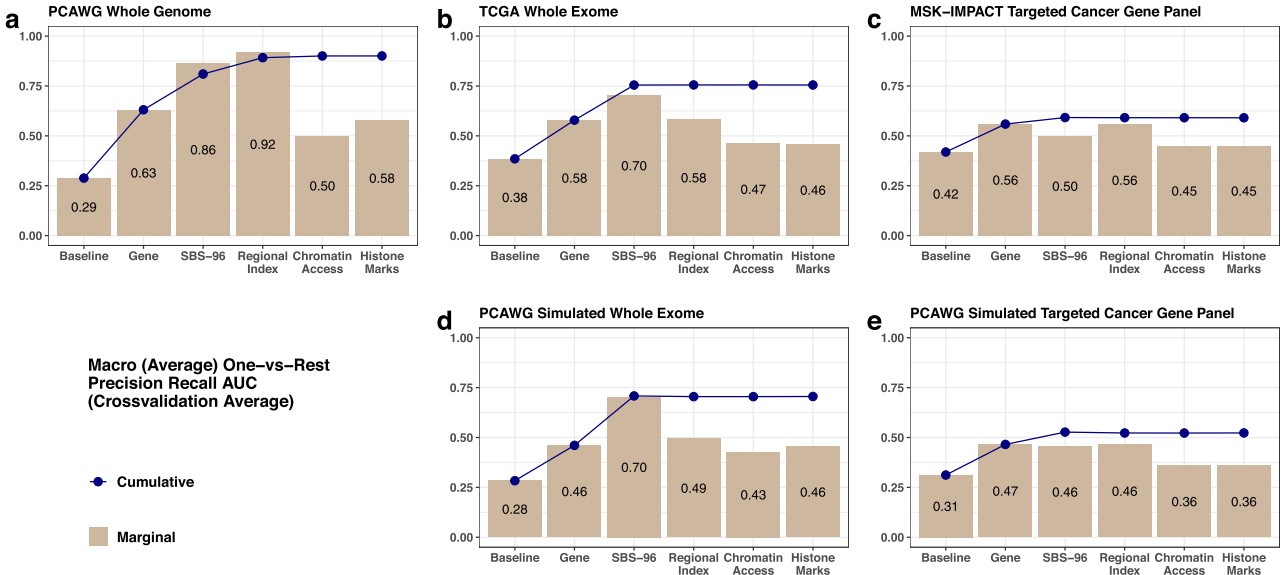

**Fig. 5 Overall marginal and cumulative effects of various meta-feature groups in the hidden genome classifier.** Bar and line charts showing the overall marginal (bars) and cumulative (lines and dots) effects of different meta-feature groups, as measured by cross-validation macro (average) one-vs-rest precision recall AUCs, in the projected hidden genome classifier for predicting cancer sites. The effects are evaluated separately in each real sequencing (PCAWG whole genome, TCGA whole exome, and MSK-IMPACT targeted cancer gene sequencing; panels—**a**, **b**, and **c**, respectively), and simulated sequencing data set (PCAWG simulated whole exome and PCAWG targeted cancer gene sequencing; panels—**d** and **e**, respectively) are shown. In each panel the leftmost bar and dot represent the macro AUC of a baseline classifier consisting of recorded variants and the square-root of total mutation burden (obtained through the trivial intercept meta-feature) as predictors. Each subsequent bar on the right shows the macro AUC of a marginal projected hidden genome classifier built with the baseline predictors and predictors associated with the corresponding meta-feature group. Each dot on the right displays the macro AUC of a cumulative projected hidden genome classifier trained with all meta-feature groups included up to that point from the left as predictors, in conjunction with the baseline predictors.

deviation increases in the predictors at their means. These quantify the effects of average individual predictors at their mean levels and can be readily compared to assess their relative importance. For this analysis we considered all meta-features in the whole genome data; however, we ignored the epigenomic meta-features in the whole exome data and ignored the 1 Mb chromosome regions as well as epigenomic meta-features in the targeted gene panel data to avoid collinearity of the predictors. Figure 6 plots the odds ratios for the top 40 predictive features with the largest absolute effects in the whole-genome dataset. These include 21 regional indices (1 Mb bin) on various chromosomes, 12 SBS-96 categories, 3 genes, and 4 individual variants. Their classification odds ratios in different cancer sites are in concordance with existing knowledge. Among the gene effects, *TP53* and *VHL* display large positive effects for classifying ovarian and kidney cancer respectively, and have mostly small effects in other categories. *KRAS* has a large odds ratio for pancreas cancer, with the specific hotspots *KRAS* G12D and *KRAS* G12R providing additional discriminative information captured by the "residual" effects at the variant level. In contrast, the influence of rare variants in the *BRAF* gene is eliminated after accounting for the discriminative effect (for skin) associated with the dominant hotspot variant *BRAF* V600E. It is worth noting that the effects from these individual genes and hotspot variants remain large in the full model, suggesting that they provide orthogonal lineage-dependency information from the mesoscale meta-features.

Among the most informative SBS-96 categories displayed in Fig. 5, T > C A.G and T > C A.T, two T > C type alcohol mutation signatures known to be associated with liver cancer, exhibit large positive effects in this disease. It is of note that some known tissue-specific mutation signatures only show substantial effects in the targeted cancer gene panel and the whole exome datasets

(Supplementary Figs. 6–9), but not in the whole genome dataset. Examples include the C > T (ultra-violet) signature in skin cancer[21]. While these signatures are observed to have large positive effects in the targeted panel and whole exome datasets they have zero estimated effects in the whole genome dataset, indicating that regional indices must be more informative predictors at the whole genome level, effectively containing the same signals as these mutation signatures.

Finally, two regional indices (the 9th and 13th Mb bin) on chromosome Y are observed to have large positive effects in classifying prostate cancer but large negative effects in breast and ovarian cancers, reflecting the obvious differences in gender among the associated cohorts. The remaining chromosome regions displayed in Fig. 6 exhibit notable tissue specificity in the PCAWG whole genome dataset, as demonstrated through the box and violin plots displayed in Supplementary Fig. 10; see Methods for more details.

**Deeper insight into the regional indices: copy number and epigenome features.** Our analyses demonstrate that selected 1 Mb indices of mutational burden carry a large proportion of the information for classifying tumor lineage. To try to understand better the relationships between mutational burden and the various epigenetic features in the model we performed an analysis of averaged values of these features at the chromosome arm level. We obtained the average normalized total mutational burden in each chromosome arm from the PCAWG whole genome dataset and performed linear regression analyses of these values in a model that included chromosome averages of each epigenomic feature (H3K35me3, H3K4me1, chromatin accessibility) obtained from the Encode and Epigenome Roadmap datasets, and included also the average copy number log ratio for the chromosome arm,

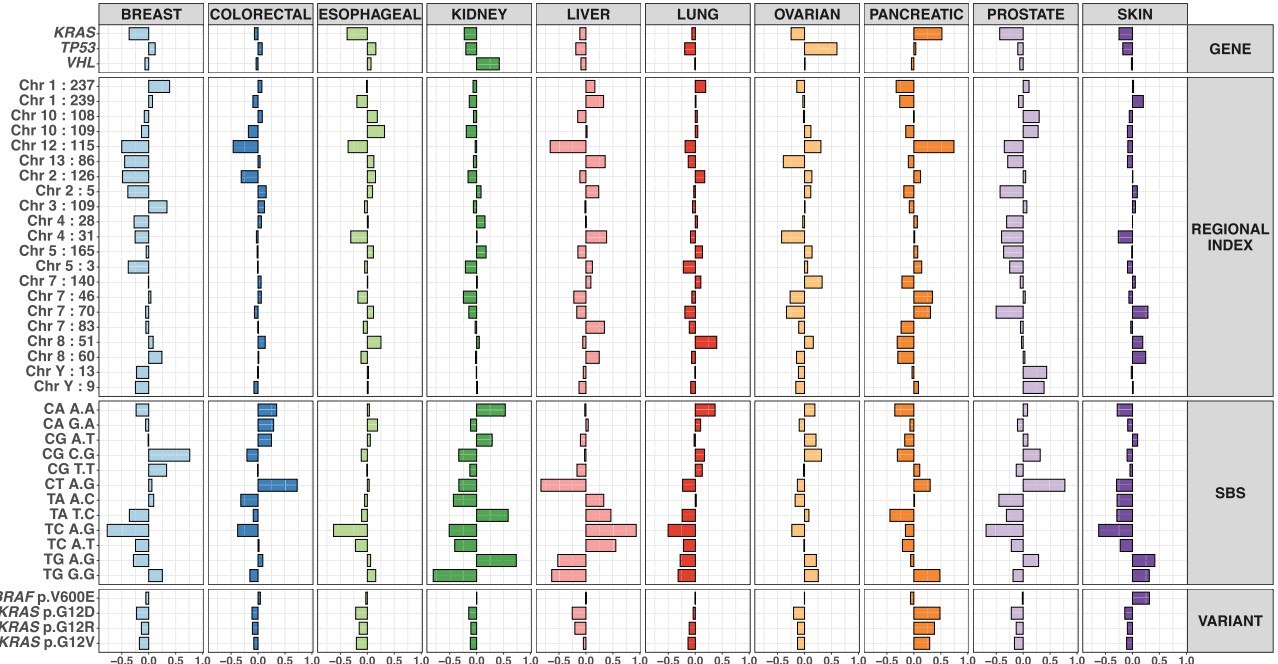

**Fig. 6 Cancer site specific odds ratios of most effective individual predictors in the hidden genome model applied to the PCAWG whole genome data.**
Log one-vs-rest odds ratios of top 40 predictors (with largest absolute log odds ratios) are first computed from the fitted hidden genome model and then plotted as bars. Each bar represents the change in the log odds of a tumor being classified into the corresponding cancer site, relative to not being classified into that site, for a one standard deviation increase in the associated predictor from its mean, while keeping all other predictors fixed at their respective means. Predictors of similar types (variants, genes, SBSs, and chromosome regional indices) are grouped together.

obtained from the TCGA whole exome copy number segmentation dataset. Individual and cumulative effects of these factors on mutational burden are displayed in Fig. 7a in terms of the percentage variance explained (*R*-squared). The patterns are strikingly different by cancer site. The dominant factor explaining mutational burden for breast and ovarian cancers is the average copy number. The strong correlations of these two factors in these diseases are shown in Fig. 7b. Similar plots for the other sites are displayed in Supplementary Fig. 11 showing more modest correlations for pancreatic and kidney and skin cancers, but negligible correlations for prostate, esophageal, and liver cancers, reflecting the regression effects in Fig. 6. In contrast, mutational burden for liver and esophageal cancers is most strongly influenced by histone marks, consistent with previous reports[28]. The strong negative correlations of mutational burden with log H3K4me1 are displayed for these sites in Fig. 7c, with plots for the other sites displayed in Supplementary Fig. 12. Supplementary Figures 13 and 14 display similar site-specific correlations of mutational burden with the other epigenomic features, viz., H3K36me3 and chromatin accessibility, respectively, and these correlations are largely similar to the corresponding correlations for H3K4me1. For many of the sites the bulk of the influence on mutational burden is largely either copy number or epigenomic meta-features, but not both. Interestingly, for skin, there seem to be substantive additive effects of both copy number and the other meta-features (Fig. 7a). It is well known that many tumor types harbor extensive copy number alterations that are cancer type dependent[26]. Our results show that the important influences of copy number appear to be captured by the RMD indices in our model.

**Comparing projected hidden genome with other classifiers**. We compared cross-validation classification performance of the proposed projected hidden genome approach with a number of black box, machine learning classifiers. We considered the following

existing approaches: (a) an SVM classifier with binary indicators of cancer gene mutations as predictors (Soh et al.[6]), (b) a random forest classifier with binary indicators of gene mutations and normalized mutation counts at all 96 SBS categories as predictors (TumorTracer; Marquard et al.[15]), (c) a deep neural network classifier, and (d) a random forest classifier with normalized mutation burdens in the ~3000 chromosome regions (RMDs) as well as in the 96 SBS categories as predictors (Jiao et al.[7]). In addition to these four, we also considered (e) an SVM classifier with the same input feature matrix as used in the full projected hidden genome approach, called the projected hidden genome SVM classifier henceforth. Note that (e) essentially produces a nonparametric generalization of the proposed projected hidden genome approach by allowing nonlinear effects of the predictors to be used in classification. A detailed note on the training strategies used for all these classifiers, including choices of loss functions used and tuning of hyperparameters performed is provided in the Supplementary Information (Supplementary Note 1).

Figure 8 summarizes one-vs-rest classification performances of all these machine learning approaches and the proposed multinomial logistic projected hidden genome classifier across all real and simulated DNA sequencing platforms considered in this study, via individual site specific and overall macro (cross-validation) precision recall AUCs. As depicted in Fig. 8 the proposed multinomial logistic classifier demonstrates competitive predictive performances across all datasets in each comparison, but in general does not have the highest AUC. Indeed, no single classifier appears to have the highest AUCs uniformly across all cancer sites in all datasets.

We note that the performances of some of these classifiers are more variable across different sequencing platforms than the projected hidden genome classifier multinomial logistic and SVM classifiers. For example, the TumorTracer[15] random forest classifier has high AUCs in whole exome and targeted cancer gene panel datasets (both real and simulated); but has

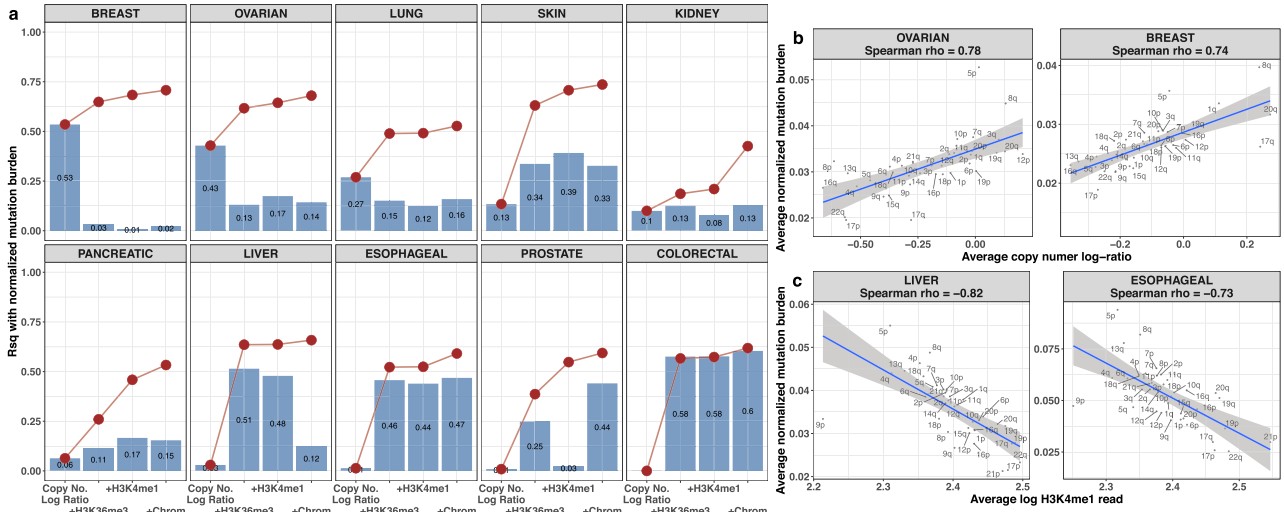

**Fig. 7 Visualizing association between normalized mutation burden and copy number alterations and epigenetic features at the chromosome arm levels. a** The marginal (bars) and cumulative (lines and dots) proportions of variability in arm level average normalized mutation burdens explained by copy number alteration log ratios and histone marks (H3K36me3 and H3K4me1) and chromatin accessibility (Chrom.), as measured through the $R^2$ statistic in the linear regression framework, are displayed for all ten cancer types. The left-most bar and dot in each subplot represent the $R^2$ for copy number alterations in explaining variability in mutation burdens. Each subsequent bar on the right shows the marginal $R^2$ of the corresponding variable, whereas and each dot represents the cumulative $R^2$ of the corresponding variable and all variables in its left in describing variability in mutation burdens. **b**, **c** Cancer site specific average normalized whole genome mutation burden in each chromosome arm is compared with the corresponding average copy number alteration log ratio (**b**) and epigenomic feature H3K4me1 (**c**) in that arm. The two cancer sites with the largest associations, as measured through Spearman's rank correlation coefficient, are displayed in each panel. The error band in each plot corresponds to 95% prediction intervals for average normalized mutation burdens obtained from a simple linear regression model fitted on the corresponding copy number log ratios (**b**) and average log H3K4me1 reads (**c**).

comparatively lower AUCs in whole genome sequencing (compared to other black box approaches). This is not surprising as our analysis shows (Fig. 5) that the groups of predictors used in TumorTracer, viz., gene and SBS-96, have relatively high marginal effects in whole exome and targeted cancer gene panel sequencing datasets, but smaller effects than the RMD (labeled regional index in Fig. 5) in the whole genome sequencing dataset. In contrast, RMD and SBS-96 are the two predictor groups with the strongest marginal effects in whole genome sequencing and near strongest effects (compared to other groups) in whole exome and targeted cancer gene panel sequencing datasets. Consequently, the deep neural network and random forest classifiers utilizing these two feature groups (Jiao et al.[7] NN and RF) are expected to achieve high accuracy in most comparisons, though the neural network classifier has poor predictive performances in the PCAWG simulated targeted cancer gene panel and TCGA whole exome datasets, potentially due to lack of convergence and/or overfitting. Finally, the projected hidden genome SVM classifier, a powerful non-parametric classifier utilizing essentially all information used in these competitors and possibly more, e.g., the individual variants, enjoys a robust performance across all comparisons in all platforms. The multinomial logistic projected hidden genome classifier, in comparison has somewhat lower AUCs.

It is worth noting in this context that the multinomial logistic classifier is not designed to solely optimize classification accuracy. The assumption of linearity in predictor effects (in the log odds scale) is potentially suboptimal for predictive modeling. Instead, the key benefit for using a multi-logistic model lies in the rigorous interpretation of predictor effects permitted by the model. In particular, the model allows formal quantification of the effect of each individual predictor (such as individual variant, gene, chromosome region, etc.) in each cancer site (see the odds ratios displayed in Fig. 6 and Supplementary Figs. 6–9). These individual effects may reveal interesting biological insights; e.g.,

our analysis quantifies the effects of known hotspot variants. Note that detection of these individual variants is accomplished de novo, and our findings concord with existing scientific knowledge. The importance of such feature effects at granular levels cannot be reliably quantified in high dimensional black box machine learning models.

Furthermore, these granular predictor effects in our model also provides a framework for coherent probabilistic quantification of modeling uncertainty. Indeed, while not done in this study, a full Bayesian implementation of the model may quantify variability both in the estimated individual predictor effects and in the predicted class probabilities, thereby facilitating rigorous statistical inference. Such uncertainty quantification and inference cannot be readily performed using black box machine learning approaches.

## Discussion

Our primary finding indicates that critical information for classifying tumor type resides in the noncoding somatic mutations detectable via whole genome sequencing. The sheer numbers of these mutations along with their relationship with discriminative epigenetic meta-features translated via local mutation burdens allow these mutations to collectively carry strong tissue-specific signals. Our method is particularly attractive because by drawing on a well-established statistical multilevel modeling framework it enables incorporation of the effects of these pan-genomic features as well as the effects of important common variants in a single model. Further, the approach permits rigorous yet straightforward evaluation of the critical factors that influence the classification. A full Bayesian implementation of our method aids quantification of estimation and prediction uncertainties.

Our results further shed light on the strong lineage dependency of the pan genomic factors that influence local mutation burdens,

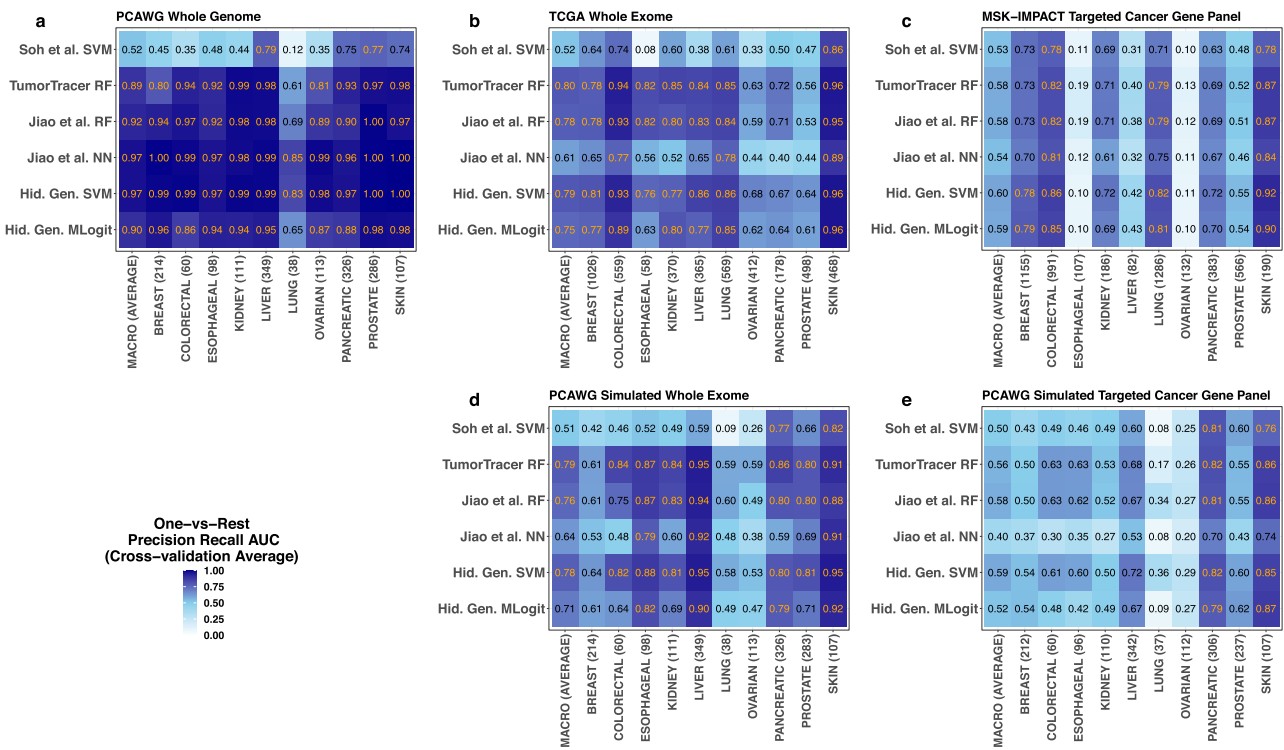

**Fig. 8 Comparing the projected hidden genome approach to other black box machine learning classifiers.** We considered the following six classifiers: (i) an SVM classifier with binary indicators of cancer gene mutations as predictors (Soh. et al. SVM), (ii) a random forest classifier with binary indicators of gene mutations and normalized mutation counts at all 96 SBS categories as predictors (TumorTracer RF), (iii) a random forest and (iv) a deep neural network classifier with normalized mutation burdens in 1 megabase chromosome regions (RMDs) and in 96 SBS categories as predictors (Jiao et al. RF and Jiao et al. NN respectively), and (v) an SVM and (vi) a multinomial logistic classifier with the full feature matrix obtained from the projected hidden genome method as predictor (Hid. Gen. SVM and Hid. Gen. Mlogit, respectively). For each of these six classifiers individual site specific and overall macro cross-validation one-vs-rest precision recall AUCs are plotted, separately for all real sequencing datasets (PCAWG whole genome, TCGA whole exome, and MSK-IMPACT targeted cancer gene sequencing; **a**, **b**, and **c**, respectively) and simulated sequencing datasets (PCAWG simulated whole exome and PCAWG targeted cancer gene sequencing; **d** and **e**, respectively). The number within parentheses on the horizontal axes labels represent sample sizes of the corresponding cancer categories.

facilitating the high classification accuracies. For some tumor sites, epigenetic features such as histone marks determine the mutation burden signal while for other sites the primary driver is copy number. Overall these relationships provide high accuracy for identifying tumor site of origin when using whole genome sequencing data. This has potential clinical value for identifying the primary site for cancers of unknown primary, and for identifying the anatomic site of tumors identified by ctDNA screening.

Our analysis is limited to somatic point mutations and it does not include other genomic alterations (such as copy number alterations), other types of omic data (such as gene expression[29] and methylation profiles), and other genomic features (such as insertions, deletions, structural variations, and whole genome doubling). If reliable information on these features is available, incorporating them into the classifier may improve prediction accuracy. However, reliable detection of many of these is difficult when tumor content is low, especially in targeted gene panel sequencing, and somatic point mutations are the only reliable features in such cases.

## Methods

**The projected hidden genome classifier: a computational overview.** We summarize the key computational steps involved in the proposed projected hidden genome classifier. A detailed note on the theoretical underpinnings of these computational steps, obtained from an efficient maximum marginal a posteriori estimation strategy of a formal and highly interpretable Bayesian hierarchical model, is provided in a following section. For each tumor $i$, consider the mutational profile vector $\boldsymbol{x}_i = \left(x_{i1}, x_{i2}, \ldots, x_{id}\right)^T$ where $d$ denotes the total number of variants observed in the test and training set combined, and $x_{ij}$ denotes the binary indicator

of the presence/absence of a variant $j$ in tumor $i$. Let $c_i \in \{1, \ldots, K\}$ be the cancer type of tumor $i$; $c_i$ is known for a training set tumor $i$, and is to be predicted for a test set tumor. Define the unit vector $\widetilde{\boldsymbol{x}}_i = \boldsymbol{x}_i / ||\boldsymbol{x}_i||$ by dividing $\boldsymbol{x}_i$ by its norm, the square-root of the total mutation burden observed in tumor $i$. For each variant $j$ let there be $p$ meta-features (dummy coded categories if discrete) with values: $\boldsymbol{u}_j = \left(u_{j1}, \ldots, u_{jp}\right)^T$. For example, if $j$ denotes the variant *KRAS* G12V and $l$ is the binary meta-feature indicating the *KRAS* gene, then $u_{jl} = 1$. We construct the meta-design matrix $\boldsymbol{U}$ by stacking $\{\boldsymbol{u}_j\}$ as rows where $\boldsymbol{U}_{\circ,l}$ is the $l$-th column of $\boldsymbol{U}$, cataloging values of the meta-feature $l$ for all variants $\{j\}$. Then the computational steps involved in the projected hidden genome classifier are as follows.

1. For each tumor $i$ and for each meta-feature $l$, compute the scalar projection of the vector $\boldsymbol{U}_{\circ,l}$ along the direction of the mutational profile of tumor $i$, viz., $\widetilde{\boldsymbol{x}}_i^T \boldsymbol{U}_{\circ,l} = \boldsymbol{x}_i^T \boldsymbol{U}_{\circ,l} / ||\boldsymbol{x}_i||$. Note that the scalar projection $\widetilde{\boldsymbol{x}}_i^T \boldsymbol{U}_{\circ,l}$ describes the total mutation burden in tumor $i$ that is attributable to the meta-feature $l$ (this is simply the total number of mutations observed in the meta-feature $l$, if $l$ is a binary category, such as the indicator of the *KRAS* gene), normalized by the square-root of the total mutation burden in tumor $i$. Note that a meta-feature vector $\boldsymbol{U}_{\circ,l}$ identically equal to 1, i.e., an intercept meta-feature (following the analogy in regression), yields $||\boldsymbol{x}_i||$, the square-root of the total mutation burden in tumor $i$ as a scalar projection.
2. Using the mutational profiles and cancer categories of training set tumors, perform a feature screening to obtain the most discriminatory $d_0$ (frequently occurring) variants, $h_1, \ldots, h_{d_0}$, where $d_0 \ll d$. In this study we used a highly efficient and scalable mutual information-based feature screening technique (see the following sections for more details) and elected to use a fixed $d_0 = 250$. After screening, obtain the entries of the unit mutational profile vector $\widetilde{\boldsymbol{x}}_i = \boldsymbol{x}_i / ||\boldsymbol{x}_i||$, associated with these most discriminatory variants $h_1, \ldots, h_{d_0}$, viz., $\{\widetilde{x}_{ih_1}, \ldots, \widetilde{x}_{ih_{d_0}}\}$.
3. Fit a group-lasso[30] penalized multinomial logistic regression model with predictors $\{\widetilde{x}_{ih_1}, \ldots, \widetilde{x}_{ih_{d_0}}\}$ and $\{\widetilde{\boldsymbol{x}}_i^T \boldsymbol{U}_{\circ,1}, \ldots, \widetilde{\boldsymbol{x}}_i^T \boldsymbol{U}_{\circ,p}\}$, and with categorical

response $\{c_i\}$ on the training set tumors $\{i\}$. The first set of predictors corresponds to indicators of discriminatory individual variants while the second set of predictors are the mutation burdens captured by the various meta-features, all normalized to the square-root of the total mutation burden encountered in the tumor. The regression parameters corresponding to the first group of predictors $\{\widetilde{x}_{ih_1}, \ldots, \widetilde{x}_{ih_{d_0}}\}$ quantify the residual effects of those individual variants, the surplus effects unexplained by the associated mutational context. In contrast, those for the second group of predictors $\{\widetilde{\boldsymbol{x}}_i^T \boldsymbol{U}_{\circ,1}, \ldots, \widetilde{\boldsymbol{x}}_i^T \boldsymbol{U}_{\circ,p}\}$ quantify the effects of the meta-features (more precisely, mutations attributed to the specific meta-features) embodying the mutational contexts.

4. For each tumor $i'$ in the test dataset, first evaluate values of the associated predictors $\{\widetilde{x}_{i'h_1}, \ldots, \widetilde{x}_{i'h_{d_0}}\}$ and $\{\widetilde{\boldsymbol{x}}_{i'}^T \boldsymbol{U}_{\circ,1}, \ldots, \widetilde{\boldsymbol{x}}_{i'}^T \boldsymbol{U}_{\circ,p}\}$ and then use these predictors with the fitted model to estimate the predictive probabilities $\{P(c_{i'} = k): k = 1, \ldots, K\}$ of the tumor $i'$ belonging to each of the $K$ cancer types. These estimated/predicted probabilities aid a soft classification of the test set tumors; a hard classification can be obtained by thresholding or assigning to the class of maximum predicted probability.

**Lower (3-) dimensional embeddings of mutation profiles**. To construct the lower three-dimensional embeddings of the classification signals from the raw mutation profile vectors $\boldsymbol{x}_i$ we first obtained the meta-feature scalar projection matrix $\widetilde{X}U$ as described above, then evaluated its first 30 principal components, and subsequently conducted a 3D t-SNE[24] on the resulting principal components to obtain lower-dimensional embeddings of the $p$-variate vector $(\widetilde{\boldsymbol{x}}_i^T \boldsymbol{U}_{\circ,1}, \ldots, \widetilde{\boldsymbol{x}}_i^T \boldsymbol{U}_{\circ,p})$ for each tumor $i$. These lower dimensional embeddings were plotted as scatterplots (a static version is displayed in Fig. 1 in the main text and an interactive html version is included within the GitHub repository for this article[25]) with each point color-coded according to its cancer site.

**The projected hidden genome classifier: a hierarchical Bayesian perspective**. The projected hidden genome classifier is built on the hierarchical Bayesian classification framework proposed in Chakraborty et al.[18], with appropriately normalized predictors to account for heterogeneity in mutation burdens across tumors. Using the same notation as introduced earlier, we consider the following multilevel multinomial logistic regression model (Eq. 1) for predicting cancer site $c_i$ based on mutation burden $\boldsymbol{x}_i$ of a tumor that utilizes the meta-feature vectors $\boldsymbol{u}_1, \ldots, \boldsymbol{u}_d$ in a hierarchical level to partially pool the raw regression coefficients associated with the variants $1, \ldots, d$ as follows:

$$P(c_i = k) = \frac{\exp\left[\alpha_k + \frac{\boldsymbol{x}_i^T}{\|\boldsymbol{x}_i\|}\boldsymbol{\beta}_{\bullet,k}\right]}{\sum_{k'=1}^{K}\exp\left[\alpha_{k'} + \frac{\boldsymbol{x}_i^T}{\|\boldsymbol{x}_i\|}\boldsymbol{\beta}_{\bullet,k'}\right]},$$
$$i = 1, \ldots, n; k = 1, \ldots, K$$
$$\beta_{j,k} \sim N\left(\boldsymbol{u}_j^T \boldsymbol{\omega}_{\bullet,k}, \tau_j^2\right), j = 1, \ldots, d; k = 1, \ldots, K \quad (1)$$
$$\omega_{l,k} \sim N\left(0, \xi_l^2\right), l = 1, \ldots, p; k = 1, \ldots, K$$
$$\gamma_k \sim N\left(0, \zeta^2\right), k = 1, \ldots, K$$
$$\tau_j^2, \xi_l^2, \zeta^2 \sim \text{Gamma}\left((K+1)/2, \lambda^2/2\right),$$
$$j = 1, \ldots, d; l = 1, \ldots, p$$

The subsequent hierarchical levels induce sparsity to the model. Writing $\boldsymbol{\beta}_{\bullet,k} = \boldsymbol{\beta}_{\bullet,k}^0 + \boldsymbol{U}\boldsymbol{\omega}_{\bullet,k}$ for all $k = 1, \ldots, K$ leads to the following mixed effect representation of the model (Eq. 2).

$$P(c_i = k) = \frac{\exp\left[\alpha_k + \frac{\boldsymbol{x}_i^T}{\|\boldsymbol{x}_i\|}\boldsymbol{\beta}_{\bullet,k}^0 + \frac{\boldsymbol{x}_i^T}{\|\boldsymbol{x}_i\|}\boldsymbol{U}\boldsymbol{\omega}_{\bullet,k}\right]}{\sum_{k'=1}^{K}\exp\left[\alpha_{k'} + \frac{\boldsymbol{x}_i^T}{\|\boldsymbol{x}_i\|}\boldsymbol{\beta}_{\bullet,k'}^0 + \frac{\boldsymbol{x}_i^T}{\|\boldsymbol{x}_i\|}\boldsymbol{U}\boldsymbol{\omega}_{\bullet,k'}\right]},$$
$$i = 1, \ldots, n; k = 1, \ldots, K$$
$$\beta_{j,k}^0 \sim N\left(0, \tau_j^2\right), j = 1, \ldots, d; k = 1, \ldots, K \quad (2)$$
$$\omega_{l,k} \sim N\left(0, \xi_l^2\right), l = 1, \ldots, p; k = 1, \ldots, K$$
$$\gamma_k \sim N\left(0, \zeta^2\right), k = 1, \ldots, K$$
$$\tau_j^2, \xi_l^2, \zeta^2 \sim \text{Gamma}\left((K+1)/2, \lambda^2/2\right),$$
$$j = 1, \ldots, d; l = 1, \ldots, p$$

In the above multilevel multinomial logistic regression model, the $\boldsymbol{U}\boldsymbol{\omega}_{\bullet,k}$ terms capture the effects of variants as explained by the associated mutation meta-features and the $\boldsymbol{\beta}^0$ terms capture the residual effects of variants unexplained by mutation contexts. We emphasize that the effects of the vast majority of the

mutations are effectively completely explained their associated mutation contexts; the residual effects are consequential only for a few highly discriminative commonly occurring variants. For implementation of the model, we adopt a maximum marginal a posteriori estimation strategy where the key parameters of interest in the model, viz., $\boldsymbol{\alpha}, \boldsymbol{\beta}^0, \boldsymbol{\omega}$, are estimated by maximizing their marginal log-posterior density:

$$\log \pi\left(\boldsymbol{\alpha}, \boldsymbol{\beta}^0, \boldsymbol{\omega}, |, \lambda, c_1, \ldots, c_n, \boldsymbol{x}_1, \ldots, \boldsymbol{x}_n\right)$$
$$= \sum_{i=1}^{n}\sum_{k=1}^{K}\mathbf{1}(c_i = k)\log\left(\frac{\exp\left[\alpha_k + \frac{\boldsymbol{x}_i^T}{\|\boldsymbol{x}_i\|}\boldsymbol{\beta}_{\bullet,k}^0 + \frac{\boldsymbol{x}_i^T}{\|\boldsymbol{x}_i\|}\boldsymbol{U}\boldsymbol{\omega}_{\bullet,k}\right]}{\sum_{k'=1}^{K}\exp\left[\alpha_{k'} + \frac{\boldsymbol{x}_i^T}{\|\boldsymbol{x}_i\|}\boldsymbol{\beta}_{\bullet,k'}^0 + \frac{\boldsymbol{x}_i^T}{\|\boldsymbol{x}_i\|}\boldsymbol{U}\boldsymbol{\omega}_{\bullet,k'}\right]}\right) \quad (3)$$
$$- \lambda\sum_{j=1}^{d}\|\boldsymbol{\beta}_{j,\bullet}^0\| - \lambda\sum_{l=1}^{p}\|\boldsymbol{\omega}_{l,\bullet}\|.$$

The above log marginal posterior density (Eq. 3) is essentially a group-lasso penalized log-likelihood of a multinomial logistic regression model with predictors $\frac{\boldsymbol{x}_i^T}{\|\boldsymbol{x}_i\|}$ and (the scalar projections) $\frac{\boldsymbol{x}_i^T}{\|\boldsymbol{x}_i\|}\boldsymbol{U}$, response $c_i$, and group lasso penalty parameter $\lambda$, and hence can be optimized using existing software[31].

**Screening individual variants using mutual information**. The scalar projections $\{\frac{\boldsymbol{x}_i^T}{\|\boldsymbol{x}_i\|}\boldsymbol{U}\}$ combine information from all (including the extremely rare) variants in the penalized multinomial logistic likelihood as presented above. Given these quantities one thus only needs to consider the most discriminative individual variants with substantial residual effects (effects not explained by meta-features), and remove all less-discriminative residual variant effects from the penalized multinomial logistic likelihood. In this study we performed a mutual information-based feature screening of variants to filter out the less discriminative entries of $\frac{\boldsymbol{x}_i}{\|\boldsymbol{x}_i\|}$ from the model prior to group-lasso estimation. To this end, for each variant $j$ we first computed the mutual information, an information theoretic measure of association/dependency, between the variant and the cancer categories using the formula:

$$\text{MI}\left(X_j, C\right) = \sum_{x=0}^{1}\sum_{k=1}^{K}P\left(X_j = x, C = k\right)\log\frac{P\left(X_j = x, C = k\right)}{P\left(X_j = x\right)P(C = k)}$$
$$= \sum_{x=0}^{1}\sum_{k=1}^{K}P\left(X_j = x|C = k\right)P(C = k)\log\frac{P\left(X_j = x|C = k\right)}{\sum_{k=1}^{K}P\left(X_j = x|C = k\right)P(C = k)}. \quad (4)$$

Here $[X_j = x]$ denotes the event of encountering variant $j$ a total of $x$ times in a tumor $(x = 0, 1)$, $[C = k]$ denotes the event that the associated tumor has cancer of type $k$ and $P(A)$ denotes the probability of an event $A$. The sample analogs of the respective probabilities (i.e., the corresponding relative frequencies) together with the convention of $0\log 0 = 0$ are used in the estimation of these mutual information values. The estimated mutual information values provide a ranking among the variants, with higher mutual information implying better discriminative ability. In this study we elected to filter out all variants falling below the rank threshold of 250. Note that, because relative frequencies can be evaluated virtually effortlessly in (even large) sparse datasets, computation of each value of mutual information is extremely efficient, and hence the resulting screening strategy is highly scalable in whole genome datasets. After screening out all but the top few variants (with mutual information rank $\leq 250$) in $\frac{\boldsymbol{x}_i^T}{\|\boldsymbol{x}_i\|}$ we carried out the group lasso estimation with the group lasso parameter appropriately tuned via cross-validation.

**One-vs-rest odds ratio from a multinomial logistic regression model**. Given the multinomial logistic regression model

$$P\left(c = k|\boldsymbol{z}_i\right) = \frac{\exp\left[\alpha_k + \boldsymbol{z}_i^T \boldsymbol{\gamma}_{\bullet,k}\right]}{\sum_{k'=1}^{K}\exp\left[\alpha_{k'} + \boldsymbol{z}_i^T \boldsymbol{\gamma}_{\bullet,k'}\right]} \quad (5)$$

interest may lie in finding the average effect of a specific predictor, say the $l$-th predictor in each cancer type $k$. Here $\boldsymbol{z}_i$ represents the vector of all predictors for the $i$-th tumor in the projected hidden genome classifier—normalized mutation indicators for discriminant variants and meta-feature scalar projections combined, and $\boldsymbol{\gamma}$ denotes the matrix of all regression parameters ($\boldsymbol{\beta}^0$ and $\boldsymbol{\omega}$ combined). A rigorous quantification of the individual predictor effects in this framework is aided by odds ratios–either relative to a prespecified baseline category $k^*$ (one-vs-one odds ratios), or relative to all other categories (one-vs-rest odds ratios). In this study we focused on one-vs-rest odds ratios for one standard deviation increase in the $l$-th predictor from its mean while keeping all other predictor variables fixed at

their respective means. This is given by

$$
\frac{P(c = k|\bar{z} + s_l e_l)}{1 - P(c = k|\bar{z} + s_l e_l)} \bigg/ \frac{P(c = k|\bar{z})}{1 - P(c = k|\bar{z})}
$$

$$
= \exp\left(s_l \gamma_{l,k}\right) \frac{\sum_{k' \neq k} \exp\left[\alpha_{k'} + \bar{z}^T \gamma_{\bullet,k'}\right]}{\sum_{k' \neq k} \exp\left[\alpha_{k'} + \bar{z}^T \gamma_{\bullet,k'} + s_l \gamma_{l,k'}\right]}. \quad (6)
$$

Here $\bar{z}$ denotes the sample mean vector of all predictor variables, $s_l$ is the sample standard deviation of the $l$-th predictor, and $e_l$ is the $l$-th unit vector, i.e., the binary 0–1 vector with $l$-th entry 1 and all other entries zero. Estimation of the above odds ratio is performed by employing estimates of $\alpha$ and $\gamma$ into the above formula.

Note that the group lasso penalized estimates of the multinomial logistic regression coefficients $\{\gamma_{j,\bullet}\}$ are either all zero or all non-zero for a given predictor (variant or meta-feature) $j$, across all cancer sites, thus aiding a predictor level selection of variables in the fitted model. In the model fitted on the whole-genome dataset only 27 (out of 245; 11%) variants, 187 (out of 604; 31%) genes, 23 (out of 96; 24%) SBSs, and 275 (out of 2915; 9%) chromosome regional indices (1 Mb bins) were selected, i.e., had non-zero regression coefficient estimates. Among those selected features the top 40 with the largest absolute effects in the whole-genome dataset were identified and their cancer site specific log odds-ratios were plotted in Fig. 5. Similar plots for the other datasets were also obtained (displayed in Supplementary Figs. 6–9).

**Summary measures for assessing classification performance**. We use the precision-recall curve and the area under the curve to assess predictive performance of a soft multi-class classifier providing probabilities for each tumor. For each tumor type we first obtain a one-vs-rest binary classification for each sample point, by adding the probabilities associated with all other tumor types for that point into a new class labeled rest. Then a precision recall curve is obtained from the resulting binary soft classification for all points, by first considering several thresholds for these probabilities to aid multiple hard classifications, and by consequently computing the precision (also called positive predictive value, defined as the proportion of samples classified to the corresponding class that truly belong to that class) and recall (also called sensitivity, defined as the proportion of samples truly belonging to the class that are correctly classified) evaluated at each hard-classification obtained at each threshold[32]. This produces a precision-recall curve for each cancer-site specific one-vs-rest comparison. We obtained the macro precision-recall curve as an overall measure of performance of the classifier by averaging all individual one-vs-rest precision-recall curves (i.e., averaging the cancer-site specific recall values corresponding to each precision in the curves). A curve with high recall for most precision values (i.e., is close to the (1, 1) "top-right" corner) indicates good classification. The area under the resulting precision-recall curve (AUC) in the unit square provides a univariate summary of classification performance, with a larger area indicating better classification. Once AUCs from each one-vs-rest classification are obtained, a univariate summary of the overall performance of the classifier across all classes can be obtained by defining the macro AUC, which is simply the unweighted arithmetic mean of the individual class specific AUCs. Note that precision recall curves and the associated areas under the curves provide better assessment of classification than receiver operator characteristic curves in multi-class problems which necessarily produce imbalanced one-vs-rest classifications when the number of classes is bigger than two[32].

We also considered one-vs-one binary classification for all cancer site pairs from the predicted multinomial probabilities. To this end, for each cancer site pair, we first obtained the corresponding conditional classification probabilities for the site pair: given predicted multinomial probabilities $p_i$ and $p_j$ for two classes $i$ and $j$, the conditional (on these two classes) predicted probabilities are $p_i / (p_i + p_j)$ and $p_j / (p_i + p_j)$ respectively. Similar precision recall analyses and evaluation of the area under the precision recall curves were subsequently obtained for all ordered paired classifications. Note that the ordering of classes in each pair is important in this analysis, as precision recall AUC is not symmetric in the positive and the negative classes. These one-vs-one precision recall AUCs are displayed as heatmaps in Fig. 4.

**Processing epigenomic feature and copy number alteration datasets**. We obtained the epigenomic data from ENCODE[22] and the Epigenome Roadmap Study[23], which provide for tumors of different cancer types the signals/reads on these features at base level positions on the chromosomes. We focused on tumors in these datasets from the same ten cancer sites considered in the main classifier: breast, colorectal, esophageal, kidney, liver, lung, skin, ovarian, pancreatic, and prostate. The data sources and cancer histologies considered in each site for the three epigenomic features, chromatin accessibility (DNAsel), H3K36me3, and H3K4me1, are summarized in Supplementary Data 1 (Sheet 4 in the excel file provided as Supplementary Data 1).

The following preprocessing steps were performed separately on each of the three epigenomic feature datasets. In each dataset, in each tumor, at each one-megabase length chromosome bin, we computed the corresponding total signals/reads above the global median signal in the dataset to reduce noise in signals; these bin-specific thresholded total reads were subsequently averaged across all tumors

from the same cancer site within each dataset. We then performed a log-log transformation (more precisely the transformation $(f(x) = \log[1 + \log(1 + x)])$ on these average thresholded total reads, and used them as meta-features in the model. Note that these collectively produce a set of 30 meta-features in the model: ten meta-features (corresponding to the ten cancer sites) each for the three epigenomic meta-features. Through scalar projections with the mutational profile of a tumor, these 30 meta-features produce a set of 30 scores that are subsequently used as predictors in the multinomial logistic model.

The data on copy number alteration log ratios were obtained for the same ten cancer sites from the TCGA segmentation dataset[33]. For each chromosome arm, as defined by cytobands, we evaluated the average (across tumors) copy number alteration log ratio of all locations belonging to that arm, separately for each cancer site. Similar summarizations over chromosome arms were performed on the processed megabase bin level epigenomic datasets, and megabase bin level normalized mutation burden (from PCAWG whole genome) datasets. These arm-level average values of copy number alterations, epigenomic features, and normalized mutation burdens are displayed on Fig. 7.

**Analyzing association between normalized mutation burden and epigenomic factors and copy number alterations for different tissue types**. To visualize the associations between normalized whole genome mutation burdens and epigenomic factors and copy numbers in a typical tumor, we first obtained the following cancer-site specific quantities:

1. The average normalized total mutation burden in each autosome arm (as defined by cytobands) for an average tumor of each tissue type in the PCAWG whole genome dataset. This is obtained by first averaging all regional mutation densities over all chromosome bins associated with the arm in each PCAWG tumor, and then by averaging these arm-specific normalized mutation burdens across all PCAWG tumors of the same cancer type.

2. The average copy number alteration log ratio in each autosome arm (as defined by cytobands) for an average tumor of each tissue type in the TCGA copy number segmentation dataset.

3. The average epigenomic feature (separately for H3K36me3 marks, H3K4me1 marks, and chromatin accessibility) in each autosome arm (as defined by cytobands) for an average tumor of each tissue type in the ENCODE and Epigenome Roadmap datasets.

We considered linear models (separately for each cancer type) with arm-level average normalized mutation burdens as responses and one or more arm-level average copy number log ratios and epigenomic features as predictors, and subsequently computed the associated $R^2$ statistics to quantify the proportions of variability in the normalized mutation burdens that are explained by these predictors (marginally as well as cumulatively) (Fig. 7a).

We also constructed scatterplots to visualize the association of autosome arm-level average normalized mutation burdens with average copy number log ratios and average epigenomic features separately for each cancer type (Fig. 7b, c and Supplementary Figs. 11–14). The degrees of association in each scatterplot was quantified by the Spearman correlation coefficient.

**Visualizing tissue-specificity of regional mutation indices**. We visualized the tissue specificity of the normalized mutation burdens (scalar projections) in the whole-genome sequencing dataset for the top 21 chromosome regions with largest absolute log odds ratios displayed in Fig. 7 via box and violin plots. To this end, we first scaled the associated scalar projections for these windows in each tumor by the corresponding tumor-specific standard deviations to aid direct comparability. Box and violin plots were subsequently constructed for each region based on the scaled normalized mutation burdens across tumors separately for each cancer type (displayed in Supplementary Fig. 10). For each region, the differences in the relative lengths and positions of the associated boxes and violins for different cancer sites demonstrate its tissue specificity.

**Reporting summary**. Further information on research design is available in the Nature Research Reporting Summary linked to this article.

## Data availability

The TCGA whole exome somatic mutation data used in this study are openly available in the GDC database [https://gdc.cancer.gov/about-data/publications/mc3-2017]. The MSK-IMPACT somatic mutation data used in this study are available in the cBioPortal database [https://www.cbioportal.org/study/summary?id=msk_impact_2017]. The controlled access PCAWG whole genome sequencing datasets are deposited at the ICGC database [https://dcc.icgc.org/]. The data is available under restricted access, access can be obtained by contacting daco@icgc.org. The exact processed subsets of the TCGA and MSK-IMPACT datasets used in our analysis are included as R data objects within the custom R package hidgenclassifier[25] developed in this study and released publicly through GitHub [https://github.com/c7rishi/hidgenclassifier]. In the same GitHub repository an interactive html version of Fig. 1 is also stored. Individual sources for the publicly available epigenomic data sets used for construction of meta-features are listed

in Sheet 4 of the excel file provided as Supplementary Data 1. The remaining data are available within the Article or Supplementary Information are available from the authors upon request.

## Code availability

An open source software implementing our methodology has been released in the public domain[25] (GitHub: https://github.com/c7rishi/hidgenclassifier). The package also contains the exact processed subsets of two publicly available datasets, viz., TCGA whole exome and MSK-IMPACT targeted gene panel datasets, that are used in our analysis.

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

## Acknowledgements

This research was supported by the following National Cancer Institute awards: R01 CA251339, P01 CA206980, and P30 CA008748. The work was performed in part at Memorial Sloan-Kettering Cancer Center, Department of Epidemiology and Biostatistics' Computing Cluster, and at the University at Buffalo's Center for Computational Research[34].

## Author contributions

S.C., C.B.B., and R.S. designed the research. S.C. made software implementations and analyzed the data. A.M. collected and processed various epigenetic datasets used in the study. Z.G. made software implementations of various deep learning-based methodologies. S.C., A.M., Z.G., C.B.B., and R.S. wrote the paper.

## Competing interests

The authors declare no competing interests.
