## [Peer Review File · Nature Communications]

REVIEWER COMMENTS

Reviewer #1 (Remarks to the Author): Expert in biostatistics

It's an interesting idea to infer the site of origin based on somatic mutations alone. After reading the manuscript with great interest, I have a few suggestions to share.

a) Carcinoma of unknown primary (CUP). First, it's good to explain in the introduction when the site of origin will be unknown and what would be the clinical consequences of it. It may sound a silly question: should the site of origin of tumor be always known? how is it possible the surgeon forgot from which organ the tumor was surgically removed? I guess the problem of site of origin would arise not for primary tumors but for metastases. Adding some words about under which situations that the site of origin is unknown (e.g., CUP, PMC2631214) would increase the practical value of the method.

b) Mutational signatures. Analysis of mutational signature is a very active research area, for which all mutations are used with no mutations discarded or hidden from the analysis. It's intuitive why mutational signatures caused by environment exposures would help infer the site of origin. For example, mutational signature of UV along could distinguish melanoma from tumors from internal organs. Indeed, authors showed SBS-96 categories as discriminatory predictors, which may be further improved using counts of mutational signatures instead of counts of SBS-96 categories.

Not all mutational signatures would be helpful for inferring the site of origin, especially when they are caused by endogenous mutational processes. For example, SBS1 due to spontaneous or enzymatic deamination of 5-methylcytosine is universal for all cancer types. Hence, it's desired to exclude mutations caused by SBS1 and other common signatures due to endogenous mutational processes in prior or by the proposed model. This is however infeasible for SBS-96 categories, which are mixed bags of mutational signatures caused by exogenous mutagenic exposures and endogenous mutational processes. It's of interest to use mutational signatures instead (at least for TCGA and PCAWG, for which mutation counts by each signature are publicly available. Pubmed: 32025018)

c) Somatic copy number. The current paper mainly used point mutations. Given a number of somatic copy number alternations are specific for tumor types. Adding somatic copy number alternations as genomic meta-features may further improve the performance of the proposed method.

d) Quantifying predictive value. AUC is used to evaluate the discrimination between cancer type A vs cancer types B+C+D+..., lumping all other cancer type together. How about the misclassification between pairs of cancer types? For example, what is the probability of the inferred site as pancreatic given the true site as esophageal. It's of interest to report the 8 by 8 matrix of misclassification probabilities.

e) Competing methods. It's necessary to compare the existing methods, not just the same model with different meta-features. For example, TumorTracer (<https://bmcmmedgenomics.biomedcentral.com/articles/10.1186/s12920-015-0130-0>), Random Forest (RF) and multi-class Deep Learning/Neural Network (DNN)-based models (<https://www.nature.com/articles/s41467-019-13825-8>) and CPEM(<https://www.nature.com/articles/s41598-019-53034-3>).

e) External validation dataset. Since the proposed method is most valuable for CUP of metastases. Validation of the selected model on independent WGS of metastases study (Pubmed: 31645765) would add great value to the paper. Applying for data at: <https://www.hartwigmedicalfoundation.nl/en/applying-for-data/>

f) Other omic data types. While adding other omic data types may be out of scope of this manuscript, it's good to discuss the pro and con of using point mutations alone instead of integrating with other omic features, such as gene expression (PMC7248358) or methylation profiles, as well as other genomic features, such as INDEL, structure variations, whole genome

doubling etc.

g) Lung and colon cancers. Why lung and colon cancers were excluded from the analysis? They are 2nd and 4th prevalent cancers in US.

Bin Zhu
DCEG, NCI

Reviewer #2 (Remarks to the Author): Expert in cancer genomics and mutational signatures

In the manuscript "Mining the Hidden Genome to Map Tumor Site of Origin", Chakraborty et al. present an approach where five meta-feature types are used to construct (as termed by the authors) a "hidden genome" and, based on projection-based statistical methods, use this "hidden genome" to identify the origins of eight primary cancers.

I find this manuscript to be truly unexciting. Do not get me wrong – I quite like the topic and, indeed, previous papers on cell-of-origin were showing biologically relevant results with implications for treatment and/or understanding cancer biology. This manuscript seems more of a methodological exercise. This is also perfectly fine but methods need to be useful and, as far as I can tell, this one does not seem to be useful. Essentially, the authors construct a model (more comments on the methodology below), apply it to several cancer types, show that they can separate/predict these cancer types, and evaluate the features that allow this prediction. I cannot see any new biology or any use for this approach in the clinic. The manuscript ends with: "This has potential clinical value for identifying the primary site for cancers of unknown primary, and for identifying the anatomic site of tumors identified by ctDNA screening." It should be noted that neither identifying cancers of unknown primary nor anatomic site of tumors identified by ctDNA screening is supported by the presented results.

Unless I am missing something, there is also a tautology in this paper. To predict the tissue of origin, the authors are using meta-features that are derived by using epigenomic data from ENCODE and Epigenome Roadmap Study. These epigenomic features require prior knowledge of the tissue of origin as the authors derive the features by calculating "an average tumor of each tissue type in the ENCODE and Epigenome Roadmap datasets."

In addition to finding the manuscript unexciting and the potential tautology, I do have a number of concerns related to the applied methodology.

1. Cancer sites

It is unclear why only "8 common cancer sites" were selected. I worry that these might have been done because the method did not work for the other 20+ cancer sites. Can the authors show how the approach works on the remaining tissue types?

Further, there seems to be some confusion (or lack of information) in regard to "8 common cancer sites" as they include "breast, esophageal, kidney, liver, melanoma, ovarian, pancreatic, and prostate." First, probably melanoma should be skin if they want to list sites/tissues. Second, can they be more specific about the cancer type. For example, esophageal squamous cell carcinoma is very different from esophageal adenocarcinoma. Similarly, kidney can be sub-classified in a number of different cancer types; is there model predicting all kidney cancer types as one (i.e., clear cell, papillary, and chromophobe) or these are separated. Again, they should really include all cancer types at least for data derived from PCAWG and TCGA.

2. Constructing the meta-features

There is simply not enough information in the methods or supplementary materials to know whether the meta-features are constructed correctly. Specifically:

- How were "SBS-96: the single-base substitution signatures" derived? In most cases, base-substitution signatures cannot be derived from gene panels (such as the used MSK-IMPACT dataset) and work only for some hypermutated samples. Similarly, mutational signatures are not accurate for exome data with less than 1 mutation per MB.
- How were regional densities of mutation burden calculated? Especially for exome and panel data. Or were these meta-features ignored for these types of data?
- Importantly, the authors used chromatin accessibility and histone modification as their features by matching them to tissues in ENCODE and Epigenome Roadmap Study. The authors should provide supplementary tables showing how these were matched? Were cell lines or normal tissues use? Which tissues were matched with which cancer types? Only breast is made up of at least four tissues: lobules, ducts, fatty connective tissue, and fibrous connective tissue. ENCODE has generate data for a number of these and they can be matched differently. For example, were lobules matched with breast lobular carcinoma and ducts with breast invasive ductal carcinoma?

3. Imbalance of different cancer types in the dataset

It is hard to evaluate the ROC and actual prediction power of the model as the authors likely have large imbalances in the numbers of samples from different cancer types that may be biasing the results. The authors should provide Matthews correlation coefficients and discuss their predictive power for each tissue type separately.

4. Computational tool

If the main advance in this paper is a novel classifier, the authors should develop an easy to use tool and explain why researchers/clinicians/etc. might want to use their classifier as well as what data is needed for the classifier to work.

Reviewer #1 (Remarks to the Author): Expert in biostatistics

It's an interesting idea to infer the site of origin based on somatic mutations alone. After reading the manuscript with great interest, I have a few suggestions to share.

a) Carcinoma of unknown primary (CUP). First, it's good to explain in the introduction when the site of origin will be unknown and what would be the clinical consequences of it. It may sound a silly question: should the site of origin of tumor be always known? how is it possible the surgeon forgot from which organ the tumor was surgically removed? I guess the problem of site of origin would arise not for primary tumors but for metastases. Adding some words about under which situations that the site of origin is unknown (e.g., CUP, PMC2631214) would increase the practical value of the method.

Response: Cancer of unknown primary typically presents at metastatic stage where the primary tumor has already metastasized and spread to other organ sites. At the time of CUP diagnosis, there may be one or more lesions present but the primary site is unknown. We have added a note in the Introduction to explain this. (p. 1, lines 35 – 40, highlighted in red.)

b) Mutational signatures. Analysis of mutational signature is a very active research area, for which all mutations are used with no mutations discarded or hidden from the analysis. It's intuitive why mutational signatures caused by environment exposures would help infer the site of origin. For example, mutational signature of UV along could distinguish melanoma from tumors from internal organs. Indeed, authors showed SBS-96 categories as discriminatory predictors, which may be further improved using counts of mutational signatures instead of counts of SBS-96 categories.

Not all mutational signatures would be helpful for inferring the site of origin, especially when they are caused by endogenous mutational processes. For example, SBS1 due to spontaneous or enzymatic deamination of 5-methylcytosine is universal for all cancer types. Hence, it's desired to exclude mutations caused by SBS1 and other common signatures due to endogenous mutational processes in prior or by the proposed model. This is however infeasible for SBS-96 categories, which are mixed bags of mutational signatures caused by exogenous mutagenic exposures and endogenous mutational processes. It's of interest to use mutational signatures instead (at least for TCGA and PCAWG, for which mutation counts by each signature are publicly available. Pubmed: 32025018)

Response: We first note that unlike the single base substitution signature (SBS) categories, which are automatically determined given single nucleotide alterations detected in any sequencing platform, determination of the interpreted/derived mutation signature categories require computation of mutation burdens attributable to these 96 SBS categories, which are then compared with precomputed weights from a reference database^{1,2}. So far these reference weights have only been derived for whole genome and whole exome mutation burdens, and determination of the derived mutation signature category entirely from targeted cancer gene panel mutations is difficult. Instead, by using the raw SBS categories our model avoids this issue

and derives de-novo weights for the 96 SBS categories through the penalized multi-logit regression. This permits construction of a unified framework that can be used to compare results from different levels of sequencing. In addition, the raw SBS-96 categories, which are characteristics of a *variant* describing the *mutation context* (i.e., are meta-features), can be directly incorporated into the proposed multilevel modeling framework to quantify effects of individual rare and unseen variants (see Supplementary text). We have added a note contrasting SBS-96 categories from interpreted mutation signatures on lines 166 – 174 on p. 6 of the revised manuscript (highlighted in red).

Second, in the proposed Projected Hidden Genome classifier, the embedded group-lasso penalty facilitates detection of *predictive* signatures, an appropriate combination/subset of SBS-96 categories that are truly predictive of specific cancer types (i.e., aids variable selection), under reasonable model justifications. This implies that if a signature lacks tissue specificity, such as the one the reviewer pointed out, the corresponding estimated regression coefficients across cancer sites will likely be all zero (or small). Consequently, the signature will have limited, if any, contribution to prediction. A note on variable selection through group-lasso penalty is added on lines 131 – 135 on p. 5 (in red) of the revised manuscript.

c) Somatic copy number. The current paper mainly used point mutations. Given a number of somatic copy number alternations are specific for tumor types. Adding somatic copy number alternations as genomic meta-features may further improve the performance of the proposed method.

Response: Copy number detection (particularly low-level chromosomal gains and losses) requires a much higher tumor content than somatic point mutations. In scenarios where tumor content is extremely low (this usually happens in a fraction of tumor samples as well as in plasma sequencing context), somatic mutations are typically the only information that is available for diagnostic purposes. Therefore, we focus on the mutation only model to assess the diagnostic accuracy for cancer types in the current study. We have addressed this in the revised Introduction (p. 4, lines 90 – 98) and Discussion (p. 16, lines 504 – 511).

d) Quantifying predictive value. AUC is used to evaluate the discrimination between cancer type A vs cancer types B+C+D+..., lumping all other cancer type together. How about the misclassification between pairs of cancer types? For example, what is the probability of the inferred site as pancreatic given the true site as esophageal. It's of interest to report the 8 by 8 matrix of misclassification probabilities.

Response: Thank you for the suggestion. In the revised manuscript we have added results on predictive accuracies from all pairwise comparisons (lines 292 – 302 on p. 10; Figure 4).

e) Competing methods. It's necessary to compare the existing methods, not just the same model with different meta-features. For example, TumorTracer

(<https://bmcmedgenomics.biomedcentral.com/articles/10.1186/s12920-015-0130-0>), Random Forest (RF) and multi-class Deep Learning/Neural Network (DNN)-based models (<https://www.nature.com/articles/s41467-019-13825-8>) and CPEM(<https://www.nature.com/articles/s41598-019-53034-3>).

Response: In the revision we have extensively compared (in an entirely new section entitled “Comparing the Projected Hidden Genome with Other Classifiers”) our approach to several other *black-box* machine learning approaches built on Random Forest, SVM and Deep Neural Network methodologies. Some of these competing methods have been previously considered in the literature³⁻⁵, and some are obtained by using the design/predictor matrix produced by our projected hidden genome model as input in a blackbox machine learning classifier. Our results (Figure 8) suggest that despite being built upon an obviously restrictive parametric assumption (linearity in the log-odds scale), the multinomial logistic model demonstrates a competitive predictive performance. The small sacrifice in predictive accuracy in our opinion is more than compensated by the interpretability of the model parameters. In particular, as demonstrated through the odds ratios displayed in Figure 6, the multinomial logistic regression model aids rigorous and coherent quantification of effects of each individual predictor (e.g., individual variants, genes, SBS-96 categories, etc.) across all cancer sites in the model, thus potentially permitting interesting biological insights at a highly granular level that is virtually impossible to obtain from a black-box machine learning method. A note on this interpretation vs. accuracy contrast is provided on p. 15 (lines 461 – 472), and a brief note on the interpretability of the model is also provided in the Introduction (p. 3, lines 80 – 86).

f) External validation dataset. Since the proposed method is most valuable for CUP of metastases. Validation of the selected model on independent WGS of metastases study (Pubmed: 31645765) would add great value to the paper. Applying for data at: <https://www.hartwigmedicalfoundation.nl/en/applying-for-data/>

Response: We should mention that the 10,000 samples from our MSK-IMPACT cohort included in the study are predominantly metastatic samples which facilitate the assessment of the method for metastatic disease. We thank the reviewer for pointing out the data source for WGS. Since this data source is non-US based, an application for access will likely take a long time with potential uncertainties.

g) Other omic data types. While adding other omic data types may be out of scope of this manuscript, it's good to discuss the pro and con of using point mutations alone instead of integrating with other omic features, such as gene expression (PMC7248358) or methylation profiles, as well as other genomic features, such as INDEL, structure variations, whole genome doubling etc.

Response: As we discussed earlier, the focus on mutation data alone is a somewhat conscious decision toward a classification model that will be feasible in applications where tumor content

is very low. Detection of these other variations are not practically feasible. We have noted these in the Discussion (p. 15, lines 504 – 511).

h) Lung and colon cancers. Why lung and colon cancers were excluded from the analysis? They are 2nd and 4th prevalent cancers in US.

Response: The ICGC-PCAWG data on these two cancer types were not available at the time of our original submission because the PDC data science cloud that hosts these data were undergoing renovation and inaccessible at the time. We have added these two cancer sites in the revised study.

Bin Zhu
DCEG, NCI

Reviewer #2 (Remarks to the Author): Expert in cancer genomics and mutational signatures

In the manuscript “Mining the Hidden Genome to Map Tumor Site of Origin”, Chakraborty et al. present an approach where five meta-feature types are used to construct (as termed by the authors) a “hidden genome” and, based on projection-based statistical methods, use this “hidden genome” to identify the origins of eight primary cancers.

I find this manuscript to be truly unexciting. Do not get me wrong – I quite like the topic and, indeed, previous papers on cell-of-origin were showing biologically relevant results with implications for treatment and/or understanding cancer biology. This manuscript seems more of a methodological exercise. This is also perfectly fine but methods need to be useful and, as far as I can tell, this one does not seem to be useful. Essentially, the authors construct a model (more comments on the methodology below), apply it to several cancer types, show that they can separate/predict these cancer types, and evaluate the features that allow this prediction. I cannot see any new biology or any use for this approach in the clinic. The manuscript ends with: “This has potential clinical value for identifying the primary site for cancers of unknown primary, and for identifying the anatomic site of tumors identified by ctDNA screening.” It should be noted that neither identifying cancers of unknown primary nor anatomic site of tumors identified by ctDNA screening is supported by the presented results.

Response: We note on the outset that this is primarily a methodological study that aims at proposing, demonstrating and exemplifying to a scientific audience a unified, general, and highly-interpretable framework for a mutation-based and meta-feature-informed approach to classification/prediction of unknown cancer sites origin that is applicable in all DNA sequencing platforms. We believe that the key contribution of our paper is methodological; however, there are indeed novel biological insights revealed by our study.

A key biological insight is our demonstration of highly discriminative diagnostic information in the non-coding regions of the genome. Our methodology provides a general and highly

interpretable framework for identification and utilization of this information. Through the use of meta-features information in extremely rare non-coding variants, comprising the vast majority of somatic alterations encountered in sequencing studies, is condensed and subsequently used in prediction. The substantial increases in diagnostic accuracy achieved by moving from panel sequencing to whole exome to whole genome data (Figure 1 and 3) within the same modeling/classification framework validates the premise that the non-coding regions of the genome crucial diagnostic information. That is, this progression largely amounts to harnessing information from rare variants in the hidden non-coding regions of the genome, since the hotspots and known cancer genes are typically already contained in the panel sequencing data. This is an important, generalized message about the location of clinically relevant information in the genome.

Again, we note that our study is primarily methodological, and the potential clinical applications, including cfDNA applications, are mentioned as motivation for the methodology.

Demonstrating the clinical utility of the method is outside the scope of this paper. However, we note that there are separate ongoing efforts on clinical application. One project that is well underway builds on a collaboration with the hepatopancreatobiliary oncology and pathology team at MSK. We applied our hidden genome model to classify the cell of origin for ampullary adenocarcinoma samples that underwent clinical sequencing. Ampullary adenocarcinoma is typically classified into two major histologic types: intestinal and pancreatobiliary. Such subtype classification based on cellular morphology has not proven to be clinically useful. However, our preliminary results (in a separate manuscript currently in preparation) have shown for the first time that the genomic classification we developed is not only prognostic, but also allows the identification of site-specific somatic alterations as potential therapeutic targets. Overall, our approach is more informative and clinically useful than the standard practice in classifying the tissue site of origin toward guiding site-specific treatment decisions.

In the revision we have clarified and emphasized the scope, motivation and contribution of the paper (p. 3 – 4, lines 70 - 98).

Unless I am missing something, there is also a tautology in this paper. To predict the tissue of origin, the authors are using meta-features that are derived by using epigenomic data from ENCODE and Epigenome Roadmap Study. These epigenomic features require prior knowledge of the tissue of origin as the authors derive the features by calculating “an average tumor of each tissue type in the ENCODE and Epigenome Roadmap datasets.”

Response: This comment led us to recognize the sentences that may have given the impression of a tautology. Note however, that no tautology actually arises in the method, as the meta-features *do not* require prior knowledge on the tissue of origin of a tumor whose primary site is being predicted. The cancer site specific epigenomic meta-features are *pre-computed* from *separate, independent* datasets, producing a set of 30 different pre-computed meta-features (H3K4me1, H3K6me3 and chromatin access, each obtained for the 10 different cancer sites we consider in the study) for all variants. Through scalar projections with the mutation profile vector for a given tumor, these tissue-specific epigenomic meta-features collectively produce a

30-component vector of scores (scalar projections) all of which are used as predictors in the classifier, thereby nullifying any possibility of a tautology. We have updated the discussion on p. 6-7 (lines 182-189) of the revised manuscript.

In addition to finding the manuscript unexciting and the potential tautology, I do have a number of concerns related to the applied methodology.

1. Cancer sites

It is unclear why only “8 common cancer sites” were selected. I worry that these might have been done because the method did not work for the other 20+ cancer sites. Can the authors show how the approach works on the remaining tissue types?

Further, there seems to be some confusion (or lack of information) in regard to “8 common cancer sites” as they include “breast, esophageal, kidney, liver, melanoma, ovarian, pancreatic, and prostate.” First, probably melanoma should be skin if they want to list sites/tissues. Second, can they be more specific about the cancer type. For example, esophageal squamous cell carcinoma is very different from esophageal adenocarcinoma. Similarly, kidney can be sub-classified in a number of different cancer types; is there model predicting all kidney cancer types as one (i.e., clear cell, papillary, and chromophobe) or these are separated. Again, they should really include all cancer types at least for data derived from PCAWG and TCGA.

Response: First, following the reviewer’s comment we have changed melanoma to skin, and have explicitly noted that our analysis focuses on esophageal adenocarcinoma and renal clear cell carcinomas in kidney. The histological subtypes considered for all 10 cancer sites in the three datasets are now summarized in Supplementary Tables 1 – 3 (included within the supplementary excel file). Second, we emphasize that this is primarily a methodological study that proposes and exemplifies the use of a unified and highly interpretable general statistical framework for classifying tumor sites based on mutation data, and compares the predictive accuracies of the proposed framework in different levels of DNA sequencing datasets, viz., targeted gene panels to whole exome to whole genome. The 10 cancer sites (the 8 sites used in the previous version in addition to colorectal and lung) used in our study corresponds to the intersection of cancer sites on which we were able to acquire at least a moderate number of data points ($n > 30$, the number of tumors corresponding to lung is only 38) in all three databases, namely PCAWG, TCGA and MSK-IMPACT, thus allowing a controlled experiment where results can be directly compared and benchmarked across independent cohorts. A comparison with other existing approaches clearly displays the competitiveness of our highly interpretable method.

2. Constructing the meta-features

There is simply not enough information in the methods or supplementary materials to know whether the meta-features are constructed correctly. Specifically:

- How were “SBS-96: the single-base substitution signatures” derived? In most cases, base-substitution signatures cannot be derived from gene panels (such as the used MSK-IMPACT dataset) and work only for some hypermutated samples. Similarly, mutational signatures are not accurate for exome data with less than 1 mutation per MB.

Response: First, note that the SBS-96 meta-feature used in the model are the 96 individual types of single nucleotide substitutions in the tri-nucleotide context, and NOT the Sanger mutation signatures (e.g., UV, smoking, etc.) derived from existing signature decomposition algorithms. Once a mutation is detected the corresponding SBS-96 category, considered as a meta-feature of that mutation in our method, is also simultaneously determined. For each tumor, through scalar projections along its mutation profile as discussed in the Method section of the manuscript, the normalized mutation burdens in all 96 SBS categories are then computed and subsequently used as predictors in the classifier. Thus, these 96 predictors/scalar-projections can be accurately calculated from a mutation profile vector of a tumor regardless of platform and genome coverage. A note on SBS-96 vs. interpreted signature is provided on p. 6, lines 166 – 174. Second, through these scalar projections supplied as predictors our model learns the optimal combination of the SBS-96 features that predicts tissue site through the penalized multi-logit regression. In other words, our model learns the tissue-site specific signatures (e.g., UV, smoking, etc) de novo instead of using the pre-computed Sanger mutational signature reference weights, the derivation of which are platform-dependent.

We note however that the predictive value of SBS-96, ranging from 0.50 (targeted), to 0.70 (whole-exome) to 0.86 (whole-genome), does depend on the genome coverage of the sequencing platform (Figure 5). This reflects the fact that the mutational signatures associated with specific sites are much less accurately represented in sparse targeted sequencing and whole exome sequencing data comparing to whole genome sequencing data, as the reviewer rightly pointed out.

- How were regional densities of mutation burden calculated? Especially for exome and panel data. Or were these meta-features ignored for these types of data?

Response: We have updated our explanation on p. 6, lines 175-177 of the revised manuscript. All meta-features were considered in the cross-validation based predictive analysis. However, for the estimation analysis (for the odds ratios shown in figure 6), this meta-feature, together with the epigenetic meta-features were ignored for the targeted panel data (p. 11, lines 342-346).

- Importantly, the authors used chromatin accessibility and histone modification as their features by matching them to tissues in ENCODE and Epigenome Roadmap Study. The authors should provide supplementary tables showing how these were matched? Were cell lines or normal tissues use? Which tissues were matched with which cancer types? Only breast is made up of at least four tissues: lobules, ducts, fatty connective tissue, and fibrous connective tissue. ENCODE has generate data for a number of these and they can be matched differently. For

example, were lobules matched with breast lobular carcinoma and ducts with breast invasive ductal carcinoma?

Response: Following the reviewer's suggestion, we have added a supplementary table (Supplementary Table 4; included inside the supplementary excel file) noting all epigenomic data sources in the revision.

3. Imbalance of different cancer types in the dataset

It is hard to evaluate the ROC and actual prediction power of the model as the authors likely have large imbalances in the numbers of samples from different cancer types that may be biasing the results. The authors should provide Matthews correlation coefficients and discuss their predictive power for each tissue type separately.

Response: Note that we did not use ROC, but instead used precision-recall (PR) curves and the associated area under the PR curves (PR AUC). These provide reasonable robustness against large imbalances in probability based binary categorical classification⁶ while obviating the need for determining any hard-classification thresholds. The Matthews correlation coefficient can potentially permit a desirable safeguard against large data imbalances, but it requires determination of optimal classification thresholds for estimated class probabilities. Because we primarily focused on soft (probabilistic) classification throughout the study instead of hard classification, we elected to use PR AUC instead of Matthews correlation coefficient. Note that, in the revision we have provided precision recall AUCs for individual site specific one-vs-rest classification, as well as paired site specific one-vs-one comparisons.

4. Computational tool

If the main advance in this paper is a novel classifier, the authors should develop an easy to use tool and explain why researchers/clinicians/etc. might want to use their classifier as well as what data is needed for the classifier to work.

Response: Thank you for the suggestion. We have created a publicly available open source software package implementing our primary methodology, and various machine learning extensions thereof, for the use of practitioners.

References

1. Alexandrov, L. B. *et al.* The repertoire of mutational signatures in human cancer. *Nature* **578**, 94–101 (2020).
2. Alexandrov, L. B. *et al.* Signatures of mutational processes in human cancer. *Nature* (2013) doi:10.1038/nature12477.
3. Soh, K. P., Szczurek, E., Sakoparnig, T. & Beerenwinkel, N. Predicting cancer type from tumour DNA signatures. *Genome Med.* (2017) doi:10.1186/s13073-017-0493-2.
4. Marquard, A. M. *et al.* TumorTracer: A method to identify the tissue of origin from the somatic mutations of a tumor specimen. *BMC Med. Genomics* **8**, 58 (2015).
5. Jiao, W. *et al.* A deep learning system accurately classifies primary and metastatic

- cancers using passenger mutation patterns. *Nat. Commun.* **11**, 1–12 (2020).
6. Saito, T. & Rehmsmeier, M. Precrec: Fast and accurate precision-recall and ROC curve calculations in R. *Bioinformatics* (2017) doi:10.1093/bioinformatics/btw570.

Thank you for your consideration. We look forward to hearing from you.

Sincerely,

Saptarshi Chakraborty¹

Colin B. Begg²

Ronglai Shen²

¹Department of Biostatistics, State University of New York at Buffalo, Buffalo, NY 14214, USA

²Department of Epidemiology and Biostatistics, Memorial Sloan Kettering Cancer Center, New York, NY 10017, USA

REVIEWER COMMENTS

Reviewer #1 (Remarks to the Author):

Most of the review comments have been addressed and I have a few minor comments.

- 1) For few figures (e.g., Fig3a and 3b, Fig8c), it's curious that the AUCs are much less than 0.5 (the benchmark when making a random guess). Are there meta-features that predict well in the training set but poorly in the validation set in these cases?
- 2) Please mention in the supplementary material file that Supplementary Figure 1 is an interactive figure as a separated file.
- 3) Few typos: "in both" to "both in" line 477. "classsifier" to "classifiers" in Fig8 legend.

Reviewer #2 (Remarks to the Author):

The authors have addressed my concerns about the overall methodology in the manuscript. They have also provided additional information and clarified some of the prior technical description. Additionally, they have provided their method as a GitHub repository. Overall, I do not have any methodological concerns in regard to this paper.

Nevertheless, I am still perplexed about the actual value of the proposed methodology. I simply do not understand when this methodology will be used in a clinical or research setting. Since this is "primarily a methodological study", can the authors include in the manuscript an example for a cancer type where their approach reveals a novel biological or clinical insight? I do not agree with their response that a "key biological insight is our demonstration of highly discriminative diagnostic information in the non-coding regions of the genome" as I do not think this is convincingly shown in the manuscript.

Essentially, when would a researcher or a clinician use their approach? I did go through the GitHub page of the method; however, the provided vignette does not provide sufficient information to answer this question. It should be noted that, in their response, the authors describe another study of ampullary adenocarcinoma where their approach is both prognostic and identifies "site-specific somatic alterations as potential therapeutic targets". Including such type of data in the current manuscript will be useful as it will provide evidence for the usability of the methodology. However, ampullary adenocarcinoma is a very rare cancer type; do the authors anticipate their method being useful in common cancer types? Can they provide another example for the current manuscript?

In summary, I do not have any technical concerns. However, I cannot recommend this manuscript for acceptance as I do not understand who or why will be using the described approach. It is entirely possible that I do not see (or misunderstand) the usability of the method. I suggest that the editor seeks additional cancer biology or clinical reviewer(s) to clarify whether the approach will be useful in their research or clinical duties.

Reviewer #1 (Remarks to the Author)

Most of the review comments have been addressed and I have a few minor comments.

1) For few figures (e.g., Fig3a and 3b, Fig8c), it's curious that the AUCs are much less than 0.5 (the benchmark when making a random guess). Are there meta-features that predict well in the training set but poorly in the validation set in these cases?

Response: This comment helped us identify an improvement to the existing figures, particularly Figures 3a and 3b, supported by additional discussion of the prediction accuracy metric used in the study, namely the precision-recall AUC. We note that unlike the receiver-operator characteristic (ROC) AUC, the precision-recall AUC does not have a fixed null baseline value of 0.5 for random guesses. Instead, for each individual site-specific one-vs-rest comparison the null baseline is the relative sample size for the site¹ (i.e., $\frac{\text{sample size for the site}}{\text{total sample size across all sites}}$); for the “overall macro” comparison the null baseline, obtained by averaging all individual site-specific null baselines, is simply $\frac{1}{\text{number of cancer sites}}$ which is 1/10 in our study. In this revision, we have displayed the null baseline values in Figures 3a and 3b as darkened areas on the corresponding precision-recall AUC bars. With reference to these baseline values, all “observed” precision-recall AUCs indicate clear positive (often substantial) improvements in predictive accuracy for the proposed classifier over “null” random guesses. In the revised manuscript we have added a short note discussing the null baseline for a precision-recall AUC on p.9 (lines 269-279).

2) Please mention in the supplementary material file that Supplementary Figure 1 is an interactive figure as a separated file.

Response: In the revision we have noted this both on the main text (p.7 line 208) and on the supplementary material (p. 1 and p. 4; highlighted in red).

3) Few typos: “in both” to “both in” line 477. "classsifier"to "classifiers" in Fig8 legend.

Response: Thank you bringing these to our attention. We have fixed these typos.

Reviewer #2 (Remarks to the Author)

The authors have addressed my concerns about the overall methodology in the manuscript. They have also provided additional information and clarified some of the prior technical description. Additionally, they have provided their method as a GitHub repository. Overall, I do not have any methodological concerns in regard to this paper.

Nevertheless, I am still perplexed about the actually value of the proposed methodology. I simply do not understand when this methodology will be used in a clinical or research setting. Since this is “primarily a methodological study”, can the authors include in the manuscript an example for a cancer type where their approach reveals a novel biological or clinical insight? I do not agree with their response that a “key biological insight is our demonstration of highly discriminative diagnostic information in the non-coding regions of the genome” as I do not think this is convincingly shown in the manuscript.

Essentially, when would a researcher or a clinician use their approach? I did go through the GitHub page of the method; however, the provided vignette does not provide sufficient information to answer this question. It should be noted that, in their response, the authors describe another study of ampullary adenocarcinoma where their approach is both prognostic and identifies “site-specific somatic alterations as potential therapeutic targets”. Including such type of data in the current manuscript will be useful as it will provide evidence for the usability of the methodology. However, ampullary adenocarcinoma is a very rare cancer type; do the authors anticipate their method being useful in common cancer types? Can they provide

another example for the current manuscript?

In summary, I do not have any technical concerns. However, I cannot recommend this manuscript for acceptance as I do not understand who or why will be using the described approach. It is entirely possible that I do not see (or misunderstand) the usability of the method. I suggest that the editor seeks additional cancer biology or clinical reviewer(s) to clarify whether the approach will be useful in their research or clinical duties.

Response:

We do believe that our analysis and results demonstrate the existence of strong discriminative information in the non-coding region of the genome that can only be fully harnessed in whole genome sequencing studies. We have further highlighted this conclusion with reference to the results from two cancer sites that demonstrate this most clearly, namely, ovarian and prostate, wherein substantial gains in discriminative signals are observed as a result of moving from targeted cancer gene panel to whole exome to whole genome sequencing dataset (in both “real” and “simulated” settings; p. 10, lines 287-289). These findings demonstrate that our multilevel statistical model is capable of harvesting effectively this important discriminative information in the non-coding genome while permitting quantification of granular effects of individual predictors – both individual variants and meta-features describing their contexts – across different cancer sites (the odds ratios displayed in Figure 6). In our study we considered only 10 common cancer types. We limited attention to these 10 sites because we feel that there are insufficient sample sizes in the remaining cancer sites to permit a fully comprehensive classification analysis of all possible cancer sites at this time. Thus a fully encompassing clinical diagnostic tool is not possible until more data become available. We note that the most effective individual predictors associated with these 10 cancer types are concordant with the existing scientific literature.

The reviewer asked about the separate ampullary study that we alluded to in our previous response letter. In that study the *parental sites* (intestinal or pancreatobiliary) of ampullary tumors are predicted through a hidden genome model trained on a number of cancer types (including colorectal adenocarcinoma, distal cholangiocarcinoma and pancreatic adenocarcinoma). The hidden genome approach has been successful in both aiding a prognostic tool for this classification/prediction problem and in permitting identification of site-specific somatic alterations as potential therapeutic targets. However, this example is small and specialized, and thus does not have the data richness that we believe is ideal for showcasing all the features of the proposed methodology. Furthermore, the ampullary study is being conducted with investigators who had no role in developing this methodology. Thus a separate publication of the ampullary study that can cite this article is the much preferred plan.

Reference

1. Saito, T. & Rehmsmeier, M. The precision-recall plot is more informative than the ROC plot when evaluating binary classifiers on imbalanced datasets. *PLoS One* **10**, (2015).

Thank you for your consideration. We look forward to hearing from you.

Sincerely,

Saptarshi Chakraborty¹

Colin B. Begg²

Ronglai Shen²

¹Department of Biostatistics, State University of New York at Buffalo, Buffalo, NY 14214, USA

²Department of Epidemiology and Biostatistics, Memorial Sloan Kettering Cancer Center, New York, NY 10017, USA

REVIEWERS' COMMENTS

Reviewer #1 (Remarks to the Author):

My minor review comments have been fully addressed. I learned something new about the precision-recall AUC. No further comments.